

# Influence of groundwater on distribution of dwarf wedgemussels (*Alasmidonta heterodon* ) in the upper reaches of the Delaware River, northeastern USA

5   Donald O. Rosenberry[1], Martin A. Briggs[2], Emily B. Voytek[3], John W. Lane[2]

[1]U.S. Geological Survey, MS 413, Bldg. 53, DFC, Denver, CO  80225  USA
[2]U.S. Geological Survey, 11 Sherman Place, Storrs, CT  06279  USA
[3]Colorado School of Mines, 1500 Illinois Street, Golden, CO  80401  USA

*Correspondence to*: Donald O. Rosenberry (rosenber@usgs.gov)





**Abstract.** The remaining populations of the endangered dwarf wedgemussel (DWM) (*Alasmidonta heterodon*) in the upper

Delaware River, northeastern USA, were hypothesized to be located in areas of greater-than-normal groundwater discharge to

the river. We combined physical (seepage meters, monitoring wells and piezometers), thermal (fiber-optic distributed

temperature sensing, vertical bed-temperature profiling, infrared), and geophysical (electromagnetic-induction) methods at

several spatial scales to characterize known DWM habitat and explore this hypothesis. Numerous springs were observed

using visible and infrared imaging along the river banks at all three known DWM-populated areas, but not in adjacent areas

where DWM were absent. Vertical and lateral groundwater gradients were toward the river along all three DWM-populated

reaches, with median upward gradients 3 to 9 times larger than in adjacent reaches. Point-scale seepage-meter measurements

indicated that upward seepage across the riverbed was faster and more consistently upward at DWM-populated areas.

Discrete and areally distributed riverbed temperature measurements indicated numerous cold areas of groundwater discharge

during warm summer months; all were within areas populated by DWM. Electromagnetic induction measurements, which

may indicate riverbed geology, showed patterning but little correlation between bulk streambed electromagnetic conductivity

and areal distribution of DWM. Collectively, multiple lines of research provide strong evidence that DWM are located

within or directly downstream of areas of substantial focused groundwater discharge to the river. Broad scale thermal-

reconnaissance methods (e.g., infrared) may be useful in locating and protecting other currently unknown mussel

populations.

**Keywords:**
Groundwater-surface-water exchange
Hyporheic exchange
Endangered species
Distributed temperature sensing
Vertical temperature profiling
Seepage meter
Geologic control



# 1 Introduction

The sediment-water interface is an important ecotone that harbors many organisms evolved to live in this dynamic environment. Areas where groundwater discharge to rivers is focused commonly are far less dynamic, particularly with regard to temperature and saturation, and some organisms rely on this stability to survive (Hayashi and Rosenberry, 2002; Smith, 2005). Such may be the case for the endangered dwarf wedgemussel (*Alasmidonta heterodon*, family Unionidae). Formerly distributed throughout eastern North America from North Carolina to New Brunswick (Moser, 1993), this species

is now endangered. It was thought to be extirpated in the upper Delaware River until populations were found along three reaches in 2000 (Cole et al., 2008). Water is diverted from three reservoirs in the upper tributaries to New York City. As a result, persistent low-flows downstream from the dams during some summers may have contributed to the demise of formerly robust dwarf wedgemussel (DWM) populations. Given the precarious status of these three DWM populations, it is imperative to understand the processes that favor their viability.

Previous studies investigated whether the riverbed in these locations would become dry during low-flow events. Although low-flow conditions dewatered much of the riverbed, areas populated by DWM remained wetted as long as river discharge exceeded 15.8 m$^3$ s$^{-1}$ at the U.S. Geological Survey (USGS) Callicoon gage (Cole et al., 2008). Another concern was excessively warm (or cold) water temperature. Other unionid mussel species are sensitive to warm water (Galbraith et al., 2012), and DWM are particularly sensitive (Heather Galbraith, personal communication). Excessive shear stress also can

reduce mussel populations (Hardison and Layzer, 2001; Layzer and Madison, 1995). The three remaining DWM populations in the upper Delaware River are located in areas where shear stress during high river stage is smaller than reach-averaged values (Maloney et al., 2012). Groundwater discharge also may be related to the location of mussel refugia (McRae et al., 2004), perhaps especially so in the upper Delaware River where groundwater discharge may locally diminish streambed dewatering (Jeffrey Cole, personal communication). Characterizing groundwater discharge dynamics to known patches of

DWM is an important first step toward a better understanding of their preferred habitat and successful management of the species.

Quantification of exchange between groundwater and surface water is particularly difficult in fluvial settings (e.g., González-Pinzón et al., 2015) due to spatial and temporal heterogeneity and multiple scales of flow that complicate distinction between hyporheic exchange and larger-scale groundwater discharge. This is further complicated in the upper



Delaware River DWM areas due to a strong similarity between surface water and shallow groundwater major-ion chemistry and very coarse bed sediment. Multiple methods are available to investigate heterogeneity of water fluxes, largely driven by geologic variability beneath and adjacent to the river, which can impart a strong control on the surficial distribution of groundwater discharge. Point-scale physical methods, such as in-river piezometers and seepage meters (Rosenberry et al., 2008), indicate the direction and magnitude of flow across the sediment-water interface. Streambed vertical temperature

profilers can extend point-in-time measurements of water flux to month-long time series of sub-daily flux estimates using automated analytical (e.g., Irvine et al., 2015; Gordon et al., 2012) and numerical (e.g., Koch et al., 2015; Voytek et al., 2014) 1-D models. When investigating groundwater discharge with superimposed hyporheic flow, it is useful to put these point-scale measurements into a larger thermal and geological context. Thermal infrared (TIR) and fiber-optic distributed temperature sensing (FO-DTS) methods are used to collect large field of view (100's of m) or extensive longitudinal (km)

water-temperature measurements (Hare et al., 2015). TIR does not penetrate the water surface while FO-DTS measures temperature along the sediment-water interface. Hydraulic head in water-table wells near the riverbank can be compared to river-surface elevations to evaluate the potential for lateral groundwater discharge. Electromagnetic-induction methods can indicate changes in streambed geology over many km, particular in areas where stream water and groundwater are of similar electrical conductivity (Ong et al., 2010). When used in combination with point-scale methodology, a more comprehensive,

process-based understanding of DWM habitat can be discerned.

   We investigated the occurrence and distribution of groundwater discharge and related factors along three reaches of the upper Delaware River. Along each reach, we compared results where DWM were present with results where they were absent. Specifically, we pursued three main goals:

   1) Determine the spatial distribution of the rate and direction of water exchange across the sediment-water interface

related to the distribution of DWM populations.

   2) Evaluate temperature dynamics at the sediment-water interface during warm, summer low-flow periods to investigate larger-scale groundwater discharge distributions, and determine whether areas populated by DWM may serve as cold thermal refugia.

   3) Investigate the geology of the riverbed and relate groundwater-surface-water exchange to potential geologic

controls.



During the course of the investigation we discovered a relatively large spring within an area populated by DWM and studied in detail the thermal influence on adjacent and downstream water (Briggs et al., 2013). Here we expand the scope more broadly to address the three goals listed above with data collected at all three DWM-populated reaches of the Delaware River.

## 2   Study area

The three study sites containing DWM are within the 43-km reach of the upper Delaware River between Hancock and Callicoon, NY (Fig. 1). Prior to this study, sites were surveyed in 2012 by biologists familiar with DWM in the Delaware River to determine the riverbed areas currently occupied by DWM. Each site encompasses areas where DWM were found as well as similar adjacent or nearby areas where DWM have not been found. Previous studies at these same three sites investigated minimum flows and temperature stability (Cole et al., 2008) and modeled shear stress related to occurrence of DWM (Maloney et al., 2012). Site 1 extends along the right (descending) side of a mid-channel island (Fig. 2A. Site 2 extends along a straight reach of the river where a single channel exists, and is centered above and below an ephemeral stream that enters the river on the right bank, approximately separating the known DWM area ($M$) and non-mussel area ($N$) Fig. 2B). The known $M$ area at Site 3 is situated along the south side of a mid-channel island, while the $N$ area is on the north side (Fig. 2C). At Site 3, DWM were found at various times over a 10-year period along the entire reach of the channel south of the mid-channel island. However, during the 2012 field season, DWM were found only along a 200-m reach at the downstream end of this channel (Fig. 2C). Because DWM were found upstream of the $M$ reach prior to 2012, the upper portion of the channel was deemed inappropriate to serve as the $N$ reach. Therefore, the $N$ reach at Site 3 was located across the mid-river island in the river channel to the left (north) of the island where current velocity was reduced. All locations are deliberately obscured to protect the endangered animals (Fig. 2).

Discharge ($Q$) at Callicoon, the downstream end of the river reach containing the three sites, has been measured by USGS since 1975 (USGS station number 01427510; http://waterdata.usgs.gov/nwis). During 1975 through 2013, $Q$ ranged from 4078 m$^3$ s$^{-1}$ in June 2006 to 8.7 m$^3$ s$^{-1}$ in September 1997. The median $Q$ during the period of record was 45.3 m$^3$ s$^{-1}$. River discharge during site visits in 2012 was close to a normally low value of about 30 m$^3$ s$^{-1}$, but discharge was much larger than normal during a site visit in 2013, ranging from 100 to 320 m$^3$ s$^{-1}$ (Fig. 3).



## 3 Methods

### 3.1 Geomorphic Parameters

Grain-size distribution of the bed surface was determined using the Wolman (1954) pebble-count method over an approximately 100-m distance within each of the M and N reaches. River depth and flow velocity were measured at every M and N location at approximately the same time. River-surface slope was surveyed along and beyond each M and N reach; combined with measurements of water depth, this provided a reach-averaged estimate of shear stress for each M and N reach. Shields stress, a dimensionless term that relates shear stress to the size of sediments on the bed (e.g., Cronin et al., 2007), was calculated and compared to critical Shields stress to determine the likelihood that the sediment bed was mobilized based on water depths measured during site visits.

Samples for water-quality analysis were collected from piezometers installed at each M and N location at Site 2 and Site 3, from water-table monitoring wells, from several seeps along the river bank, from the large spring/seep at Site 2 (Fig. 2B), and from the river at each site. Groundwater and river chemistry were found to be universally similar; therefore, these results are not discussed in further detail further.

### 3.2 Evaluation of flow between groundwater and surface water

#### 3.2.1 Visual and infrared observations

Discharge of groundwater to the river was visually evident at all three sites. Flowing water either discharged along the bank just above the river surface or was visible as it suspended sediment just beneath the river surface. A handheld TIR camera (FLIR T620, FLIR Systems, Inc., Nashua, NH) was used to locate and measure surface-temperature anomalies related to cold groundwater seepage near the streambank. TIR data was used to quickly discern between actively flowing seeps and other bank areas that were simply wet. TIR imagery represents only the temperature at the water surface; therefore, the cameras were most useful for identifying seeps at and landward of the shoreline and unmixed plumes of groundwater that reached the river surface (e.g., Hare et al., 2015). A bucket and stopwatch were used to quantify spring/seep discharge where conditions allowed.





### 3.2.2  Lateral groundwater discharge potential

A water-table monitoring well was installed adjacent to the right (southwest) bank of the river at each site to determine

the hydraulic gradient between the water table and the river.  Wells were installed using an auger to depths beneath land

surface of 2.85 and 2.81 m at Sites 2 and 3, respectively.  At Site 1, a monitoring well could not be installed because boulders

in the bank were too large and densely distributed to auger a hole.  However, a 0.46-m-deep hole was dug by hand to below

the water table at a distance of 7.1 m from the river shoreline.  This allowed a single measurement of horizontal hydraulic

gradient at Site 1 (Fig. 2A).

Discharge of groundwater to the river was calculated using the standard Darcy equation:

$Q=KiA$                                                                                  (1),

where $Q$ is the volumetric seepage rate ($m^3 d^{-1}$), $K$ is the horizontal hydraulic conductivity of the sediment between the well

and the river shoreline ($m d^{-1}$), $i$ is the horizontal hydraulic gradient, which is the difference in head (m) between the water

level at the monitoring well and the river divided by the distance from the monitoring well to the shoreline (m), and $A$ is the

cross-sectional area ($m^2$) of a vertical plane at the river shoreline through which water must pass as groundwater discharges to

the river.

A single-well slug test  (Bouwer and Rice, 1976; Bouwer, 1989) was conducted in each monitoring well to estimate $K$.

Hydraulic head higher than the river stage indicates flow from groundwater to the river; $i$ is assigned a positive value for such

a condition but $i$ also can be negative if the groundwater head is lower than river stage.  Gradient and, therefore, $Q$ was

determined every 20 minutes during July 2012 through June 2013 at sites 2 and 3 using data provided by Solinst submersible

pressure transducers (Levelogger Edge, Junior and Barologger, Solinst Canada Ltd., Georgetown, Ontario, Canada) installed

at fixed locations in each monitoring well, and in secure locations in the riverbed (Fig. 2B, C).

### 3.2.3  Seepage meters

Seepage meters directly measure water flow across an approximately 0.25 $m^2$ portion of a sediment bed in units of

volume per area per time.  Seepage meters modified for use in flowing water (Rosenberry, 2008) were installed at five

locations along each M and N reach indicated in Fig. 2.  Locations along each M and N reach were numbered 1 through 5





with numbers increasing with distance downstream. Meters also were installed within a spring area (Briggs et al., 2013) at

175 Site 2 at locations S1 and S2 where sediments were soft, fine-grained, and markedly colder. Multiple measurements (n

ranged from 3 to 7) were made at each location, both to reduce measurement uncertainty and to determine whether there was

substantial temporal variability.

### 3.2.4 Streambed Piezometers

Similar to the riverbank monitoring wells, streambed piezometers can determine the potential for the direction of flow by

measurement of pressure within the streambed compared to the surface-water stage, but on a vertical axis. Streambed

piezometers were installed directly adjacent to seepage meters at all M and N locations, except where installations were

impossible due to buried boulders or where locations were so close together that a single piezometer could represent both

locations. Piezometers consisted of a stainless-steel pointed screen (30 mm diameter and 85 mm screened interval)

connected to 27 mm diameter galvanized pipe. Piezometers were driven to approximately 0.5 to 0.6 m depth beneath the

riverbed. Completion depth was less than 0.5 m if, after several attempts, buried cobbles or boulders prevented deeper

installation. In some locations where vertical head difference was very small, the piezometer was driven to a greater depth to

create a measured head difference greater than the measurement error. Insertion depths ranged from 0.42 to 1.15 m. In-river

piezometers can indicate rates of exchange at the sediment-water interface if a value for $K$ is measured or assumed.

However, as the seepage meters already provided a direct measure of flow across the sediment-water interface, vertical head

gradients from the piezometers were combined with seepage rates from the seepage meters to determine a calculated value

for vertical hydraulic conductivity, $K_v$, at each location.

### 3.2.5 Streambed vertical temperature profiler

Surface temperature variations propagate downward into streambed sediments due to the sum of conduction and

advection; if the conductive properties of the bed are measured or assumed, vertical advection can be determined using 1D

analytical or numerical models (Constantz et al. 2008). Thermistor dataloggers (iButton Thermochron DS1921Z, Maxim

Integrated, San Jose, CA) were installed at depths ranging from 0 to 0.4 m in 14 of the piezometers to provide temperature

profiles with depth over time. These temperature records were collected for approximately 3 to 7 days. Strong, upward





groundwater flow often reduces measurable diurnal signal penetration to less than 0.2 m (Briggs et al., 2014); therefore, at

least 1 short complementary temperature profiler designed specifically to measure upward seepage was installed within the

*M* zone at all three sites. These short profilers were constructed with 4 thermistor dataloggers (iButton Thermochron

DS1922L) positioned at depths of only 0.01, 0.04, 0.07, and 0.11 m beneath the riverbed. One such profiler was installed at

Site 1 in close proximity to an observed bankside seep, 3 profilers were installed at Site 2 adjacent to seepage meters, and 2

were installed adjacent to seepage meters at the Site 3 M reach (Fig. 4, locations indicated as 1D temperature). Temperature

records were collected for approximately 25 days.

Streambed-temperature time-series data were analyzed with the VFLUX program (Gordon et al., 2012) run in Matlab

(Mathworks, Natick, MA). Diurnal signals were extracted from field data using VFLUX and applied to the amplitude-

attenuation analytical model (as described by Hatch et al. (2006)) because this model has been shown to be reliable in

determining upward flow rates (Briggs et al., 2014). This method of analysis provides the ability to resolve temporal patterns

of vertical seepage at sub-daily time steps over the period of temperature-data collection.

### 3.3    Temperature at the sediment-water interface

A Sensornet Oryx (Sensornet House, Elstree, Hertfordshire, UK) fiber-optic distributed-temperature-sensing system

(FO-DTS) was deployed on the riverbed at Sites 2 and 3 to collect continuous temperature data in space and time along linear

cables (e.g., Selker et al., 2006). The stainless-steel reinforced fiber-optic cables were distributed across 585 m of the

streambed at Site 2 and across 944 m of the streambed at Site 3. The deployment at Site 2 (July 21-July 24, 2012)

encompassed adjacent M and N reaches, while the Site 3 installation (July 25 to July 27) only covered the M reach due to

length limitations of the cable. FO-DTS data were analyzed to identify locations of anomalously cold temperature and small

thermal variance that may correspond with focused groundwater seepage to the river (e.g., Briggs et al., 2012a), and thermal

refuge for the DWM.

FO-DTS data were collected at 4- and 10-minute intervals and calibration for thermal drift was performed using a

continuously mixed ice bath monitored dynamically by a Sensornet thermistor-type thermometer. Approximately 30 m of

cable were placed in the calibration ice bath. The standard deviation of the recorded FO-DTS temperatures in the ice bath,

determined to be 0.07 °C, was used to estimate the precision of the FO-DTS datasets. The cable on the bed was geo-





referenced by correlating survey points taken with a Nivo 5M total station (Nikon-Trimble Co., Ltd, Tokyo, JP) with meter marks printed on the cable jacket.

In addition to the spatial coverage provided by the linear FO-DTS cables, manual point ("snapshot") measurements of streambed temperature were collected at 0.05 m sediment depth using a high-precision (0.01 °C) digital thermometer

(Traceable Thermometer, Control Company, Friendswood, TX) at both M and N reaches of Sites 2 and 3, similar to the method described by Lautz and Ribaudo (2012). Discrete bed temperatures were collected over approximately 2 hours at Site 2 (n=107) on July 22, and over 2.5 hours at Site 3 on July 25 (n=149) at geo-referenced locations. Data were laterally interpolated to generate areal streambed temperature maps using the ArcMap 10 (ESRI, Redlands, CA) "nearest neighbor" method.

### 3.4    Geology of the riverbed

Bedrock and unconsolidated materials have characteristic electrical-conductivity properties that can be sensed remotely with a variety of geophysical tools. Multi-frequency electromagnetic Induction (EMI) data were used to make inferences

about underlying geologic structure of the streambed; EMI has been used previously to better constrain exchange between groundwater and surface water at landscape scales (e.g., Ong et al., 2010). These data were collected at all three sites using a portable digital, multi-frequency, electromagnetic conductivity sensor (GEM-2; Geophex, Inc.,Raleigh, NC) that measures the bulk apparent subsurface electrical conductivity or magnetic susceptibility. Variance in electrical conductivity provides information about groundwater quality (e.g., salinity) or substrate properties, such as porosity. Larger conductivity values

correspond to more conductive subsurface materials, such as shale bedrock or near-surface materials with a higher silt or clay fraction, whereas smaller conductivity values may indicate sandstone bedrock or coarser grained surficial deposits. GEM-2 can be used to estimate streambed characteristics at depths up to approximately 12 m depending on streambed composition.

Multi-frequency EMI data were collected at all three sites. A fixed land location was established at each site and visited at the beginning and end of each survey to correct for instrument drift. The instrument was suspended about 1 m above the

water surface using non-metallic PVC pipe secured inside an inflatable raft. A kayak and drogue were used to position the



raft to provide areal coverage of the riverbed. All GEM-2 land locations and surveys were geo-referenced with an on-board GPS unit.

## 4 RESULTS

### 4.1 Geomorphic parameters

Median water depths measured at Site 2 during the June 2012 field visit were 0.58 m and 0.59 m for M and N locations, respectively. Median depths at Site 3 for M and N locations were 0.41 and 0.44 m, respectively (Table 1). Median river velocities were virtually identical between M ($0.18$ m s$^{-1}$) and N ($0.17$ m s$^{-1}$) measurement locations at Site 2. However, because the N locations at Site 3 were in the wider and deeper channel north of the mid-channel island, median velocity at the N locations was nearly 4 times faster than at the M locations (Table 1). Only at locations M4 and M5 were current velocities at Site 3 approximately the same as M-location velocities at Site 2.

Reach-averaged shear stress was nearly identical at the M and N locations at Site 2, primarily because the slope of the river surface (0.00037) was the same at both reaches. The M reach at Site 3 also had virtually the same slope (Table 1). The slope at the N reach at Site 3 was nearly twice as large at 0.00065. Therefore, shear stress at Site 3, reach N, was more than double that of any of the other reaches. Shields stress (Table 1) at all reaches was well below commonly assumed critical values of 0.03 to 0.06 required for bed mobility (e.g., Shvidchenko et al., 2001).

River slope and water depths were not measured at Site 1. However, Maloney et al. (2012) indicate that water depth, current velocity, and shear stress at Site 1 are similar to Site 2 during river discharge less than about 100 m$^3$ s$^{-1}$. Cole et al. (2008) also indicate similar water depths between Sites 1 and 2 in portions of the riverbed where DWM are known to be present.

### 4.2 Groundwater-surface-water exchange

#### 4.2.1 Visual and infrared observations

Walking along the Site 1 riverbank above and below the reach where DWM have been identified revealed 10 bank-side seeps on both sides of the channel southwest of the mid-channel island. Small wetland areas of approximately 10 to 30 m$^2$ areal extent existed uphill of the seeps southwest of the channel, particularly in the area where the monitoring-well





excavation was made (Fig. 2A). These wetland areas were situated 1 to 1.5 m above the river surface and between 2 and 8 m from the riverbank and were characterized by saturated soft sediments. Seeps along the left bank (island side) of the southern channel were more distinctly located. Discharge at the seep adjacent to the northernmost extent of the DWM area in Fig. 2A

280 was sufficient to cause groundwater sapping, resulting in landward erosion of sediment along a 0.1 to 0.2 m vertical face at the shoreline. No bankside seeps were identified within 200 m upstream of the northernmost seep or downstream of the southernmost seeps in Fig. 2A, nor were any seeps identified along the north side of the island. A spring situated uphill from the river along a road cut discharged 6.4 L min$^{-1}$.

The spring area at Site 2 (Fig. 2B, Fig. 5) included two areas approximately 0.1 m in diameter and separated by about 0.5

285 m that discharged copious amounts of water and is described in detail in Briggs et al. (2013). The smaller spring discharged 12.9 L min$^{-1}$ and the larger spring discharged 76.5 L min$^{-1}$ at a nearly constant rate. Combined, they discharged nearly 129 m$^3$ d$^{-1}$. TIR imagery at Site 2 indicated water issuing from these two spring discharge points had a steady temperature of 10.8 °C. This cold, dense plume of unmixed groundwater plunged into the river within 2 m of the shoreline (Fig. 5). A small volume of discharge also originated at the mouth of the ephemeral stream that was about 40 m upriver of the large seep area

(Fig. 2B). No other bankside seeps were identified along or adjacent to this study reach nor were any observed along the northern riverbank opposite the study area.

At Site 3, the riverbank immediately southwest of the riverbed area where DWM have been located did not contain any obvious seeps, but the 10 to 15-m-wide bench immediately adjacent to the shoreline was soft and wet in areas. Two seeps were identified at the shoreline next to the mid-channel island, approximately equidistant from the two westernmost M

locations (Fig. 2). These seeps discharged water both above and below the shoreline and suspended sand where the discharge point was submerged. Five other colder seeps were located upriver along the right bank (south side) of the channel. The discharge point at all of the right-bank seeps was 0.1 to 0.2 m above the river surface. Although difficult to measure, several of the seeps were discharging at approximately 0.5 to 2 L min$^{-1}$. Seeps along the mid-channel island at Site 3 were universally warmer (22.6-25.7 °C) than those observed at other sites. Seeps on the south side of the channel at Site 3 were

not measured but they were noticeably colder than the island seeps and were similar in temperature to the Site 2 seeps. Temperature in the monitoring wells at Sites 2 and 3 averaged 11.9 and 11.4 °C, respectively, during the period when spring and seep temperatures were measured.



### 4.2.2 Lateral groundwater discharge potential

The single measurement of $i$ at Site 1 during the afternoon of 26 July 2012 resulted in a value of 0.17, indicating a large

potential for groundwater to discharge to the river. The stabilized water level in the excavation was 1.22 m above river stage.

With no value for $K$ for this location, no attempt was made to determine the rate of groundwater discharge to the river along

this reach. This measurement was made during a prolonged period of relatively steady river discharge that began on June 24

and likely represents largely steady-state conditions.

The median value for $i$ at Site 2, determined over nearly a year, was 0.08 (Fig. 6A), also quite large for sandy sediments.

Values were typically close to 0.07 during summer months and increased to about 0.10 starting in November and continuing

until mid-March. The smallest gradients, other than during gradient reversals, occurred during mid-May to mid-June when

river stage rose to a greater extent than groundwater. $K$ based on analysis of slug-test data from the Site 2 monitoring well

was $6.3 \times 10^{-2}$ m d$^{-1}$, indicative of silty sand. Assuming that the sandy sediment between the riverbed and the underlying

bedrock is 10 to 20 m thick, and assuming that all of the horizontal flow of groundwater to the river occurs through this 10 to

20 m thick vertical cross section of sand and fractured shallow sandstone before discharging to the river, groundwater

discharge to the river at Site 2 would be 0.04 to 0.07 L min$^{-1}$ per m of river reach. For the entire Site 2 reach, groundwater

discharge would be about 10 to 20 m$^3$ d$^{-1}$.

The median value for hydraulic gradient based on the monitoring well at Site 3 was 0.05 (Fig. 6C). Other than during

high-discharge events, the median value was remarkably stable during the period of record. The gradual decrease in

hydraulic gradient from late July until early September, 2012, is likely a return to hydrostatic conditions following well

installation and indicative of the low $K$ of the sediments in the well. More than a month was required for the water level

inside the well to stabilize. Slug tests indicate that $K$ is $8.5 \times 10^{-5}$ m d$^{-1}$, 3 orders of magnitude smaller than at Site 2. Making

the same assumption that groundwater discharge to the river occurs through a vertical plane at the shoreline that is 10 to 20 m

thick, groundwater discharge at Site 3 would be on the order of 3 to $6 \times 10^{-5}$ L min$^{-1}$ per m of river reach, or 0.03 to 0.06 m$^3$

day$^{-1}$ for the entire Site 3 reach. Given the numerous springs in this area, this slow, diffuse groundwater discharge clearly is

augmented by focused groundwater discharge along preferential flow paths.



Relative river stage and water level in the adjacent groundwater monitoring well are plotted in Fig. 6 for Sites 2 and 3.
Values are adjusted so river stage approximately equals the water depth at each in-river pressure transducer. As river stage

rose, the shoreline moved laterally and the distance between the shoreline and the monitoring well decreased. Calculations of
$i$ incorporated a linear interpolation of reduced horizontal distance with increasing river stage, with a minimum horizontal
distance of 0.9 m occurring at a maximum relative river stage of 3.53 m at Site 2, and a minimum horizontal distance of 0.7
m occurring at a maximum relative river stage of 2.93 m at Site 3. At both sites, maximum river stage occurred on 19
September 2012 (Fig. 6), indicating the high-stage river shoreline did not quite reach the locations of the monitoring wells.

During the 1-year period from July 2012 through June 2013, the hydraulic gradient at both Sites 2 and 3 reversed and
became negative seven times in response to a rising river stage that preceded and exceeded a corresponding rise in the water
table at the monitoring wells. This effect is displayed in Fig. 6B for the largest rise in river stage at Site 2 on September 19.
Interestingly, even though river stage 18 hours after peak river discharge was still 0.75 m higher than it was prior to the high-
stage event, hydraulic gradient had already returned to the pre-event value of 0.04.

### 4.2.3 Flow measured at seepage meters

Seepage generally was small at all but a few M and N measurement locations. Median values of seepage were upward at
8 of 10 M locations and at 6 of 10 N locations. Both M reaches had positive (upward) median values (0.43 and 3.55 cm d$^{-1}$)
and both were larger than median values for the N reaches (0.18 and -0.01 cm d$^{-1}$) (Table 1). The only reach where seepage

was consistently upward was the M reach at Site 3, where substantial upward seepage was measured at 4 of 5 locations.
Seepage was more variable at N reaches than at M reaches, particularly so if the Site 3 M5 value is excluded (Fig. 7).
Seepage at that location was much more variable due to increased turbulence where two channels merge into a single
channel. The large variability at the N reach of Site 3 (Table 1) was undoubtedly due to the larger current velocity, shear
stress, and greater hyporheic exchange there.

Seepage at locations S1 and S3 at Site 2 was only slightly faster relative to seepage measured at nearby M locations
(Table 1). Although larger rates of seepage were expected within this cold-water spring area, detailed temperature
measurements indicated that most of the seepage, and source of the cold water at the streambed, originated landward of the
shoreline (Briggs et al., 2013).





### 4.2.4 Streambed Piezometers

Vertical hydraulic gradients ($i_v$) measured at in-river piezometers were generally small and indicated the potential for

upward flow at all M locations and at 6 of 8 N locations. Where $i_v$ was negative (Site 2 N4, Site 3 N4), indicating the

potential for downward flow, $q$ also was negative, indicating downward flow (Table 1). The median of M-reach $i_v$

measurements was 0.009 at both M reaches. Median values at N reaches were 0.003 and 0.001 for Sites 2 and 3,

respectively. The piezometer installed at S1 in the spring area at Site 2 (Figs. 2B and 5A) indicated a relatively large $i_v$ of

0.017.

Calculated $K_v$ at about half of the measured locations was smaller than expected for a gravel- to cobble-bedded river,

indicating that finer-grained sediments were present between the bed surface and the well screen at some locations. Values

for $K_v$ ranged from 0.1 to 39 m d$^{-1}$ at M-reach locations and from 0.2 to 313 m d$^{-1}$ at N-reach locations. The two largest $K_v$

values were at N2 and N3 at Site 3, where the current is faster and cobbles are larger. These values, both larger than 100 m d$^{-1}$, are indicative of coarse sand or gravel. Median values of $K_v$ determined at M reaches were 0.5 and 12.5 m d$^{-1}$ at Sites 2 and

3, respectively. Median values of $K_v$ at corresponding N reaches were 1.3 and 202 m d$^{-1}$. The value for $K_v$ at location S1 in

the spring area was only 0.3 m d$^{-1}$ (Table 1), indicative of silty sand, such as was observed on the bed in this area.

### 4.2.5 Streambed Vertical Temperature Profilers

Vertical seepage rates determined with VFLUX from the thermal records collected in piezometers varied substantially

depending on which pair of thermometers was used to calculate $q$ (Table 2). At Site 2, thermal results indicated that rapid

downward seepage near the surface of the riverbed decreased with depth at locations M3 and N3, whereas seepage-meter

results indicated small (M3) to moderate (N3) upward seepage. Results at Site 3 N3 were similar to those at Site 2 N3 but

with smaller values. Only at Site 3 M5 did the thermal profiler records collected in piezometers and seepage-meter results

indicate seepage in generally the same direction. Results from thermal models and seepage-meter values were similar at M2

and also at M5 if the 30-50 paired thermometers were used (Table 2).

Seepage determined with temperature data from the shallow profilers designed to capture upward flow at Site 1 (installed

near a visible bankside seep) averaged 16 cm d$^{-1}$ and steadily decreased over the recording period (Fig. 8). Shallow profiler



seepage at Site 2 near the main spring (2 to 3 m from the S1 and S3 seepage meters) averaged 27 cm d$^{-1}$ and varied from 12

to 39 cm d$^{-1}$. The remaining profiler data from two of the three locations at Site 2, and both locations at Site 3, showed fairly

consistent fluid flux that can be described as "circumneutral" as they ranged within the expected error bounds of +/- 10 cm d$^{-1}$

associated with this method in this coarse-grained setting (Fig. 8). All four of these circumneutral plots show two downward

spikes in seepage, the latter coincident with an upward spike in river stage and discharge measured at the USGS gage in

Callicoon.

Slow seepage-flux estimates in the range of +/- 10 cm d$^{-1}$ from profilers at Sites 2 and 3 generally correspond with

nearby seepage-meter rates ranging from 0.5 to 7.0 cm d$^{-1}$. The 27 cm d$^{-1}$ value from the profiler installed near the spring

area at Site 2 was substantially larger than values of 0.56 and 2.20 cm d$^{-1}$ measured at the S1 and S3 seepage meters.

However, the profiler value was similar to seepage determined at 4 HRTS sensors installed in the spring area for a related

study that averaged from 12 to 35 cm d$^{-1}$ (Briggs et al., 2013).

### 4.3 Temperature at the sediment-water interface

Average FO-DTS temperatures collected over 4 days at Site 2 ranged from 14.0 to 22.5 °C (Fig. 9). A slightly colder

zone was detected along a 115 m length of cable located closer to shore along much of the M reach and into the southern

portion of the N reach. Discharge of cold groundwater should result in decreased variance but temperature variance along the

Site 2 FO-DTS cable is actually largest for the colder areas, except at the bankside seep depicted in Fig. 5 (Fig. 9B). The

larger variance is likely due to the nearshore cable along the cold reach being situated in shallow, clear water that often

results in solar heating of the interface and cables (Neilson et al., 2010). This is supported by data from the cable located

further from shore that shows reduced thermal variance. Average FO-DTS data collected over 2 days at the Site 3 M reach

have a much tighter range of 22.6 to 23.3 °C. Temperature variance (Fig. 9D) is relatively large throughout the shallow-

water area south of the island point bar, but is greatly reduced where the cable passes through stronger current from the

channel that originates on the north side of the mid-river island.

Temperatures measured with the snapshot streambed thermal surveys at Sites 2 and 3 are generally similar to

patterns shown in the FO-DTS data, although several discrete cold zones near the island at Site 3 were detected with the bed

survey but missed with nearby FO-DTS cables. Discrete cold patches were found at Sites 2 and 3 along the M zones but not



in the N zones (Fig. 9). However, the cold anomalies make up a relatively small percentage of the overall surveyed area at both M reaches. The largest cold anomaly is located at the Site 2 spring area and indicates a plunging plume of cold water, as discussed above. The areal extent of this anomaly is approximately twice as large as the plume footprint as measured within the water column, likely indicating an influence from more diffuse groundwater upwelling through the streambed, as detailed in Briggs et al. (2013). Cold riverbed areas were better detected with the discrete snapshot method than with the continuous FO-DTS method, likely because the snapshot measurements were made at 0.05 m depth and the fiber-optic cable was resting on top of the bed and influenced to a greater extent by surface-water temperatures. The snapshot method also provided better lateral distribution of data collection.

## 4.4 Geology of the riverbed

Consistent spatial patterns of streambed electrical conductivity were observed in multiple adjacent and overlapping EMI lines, but there was no apparent relation between riverbed electrical conductivity and occurrence of DWM (Fig. 4). For example, DWM areas at Site 1 and Site 3 are located above more conductive material, whereas corresponding N reaches are generally less conductive. Conversely, DWM at Site 2 are found over the least-conductive material while the opposite side of the river and N reach are both more conductive.

## 5 DISCUSSION

Multiple methods for characterizing rates and spatial distribution of groundwater discharge collectively lead to the conclusion that seepage is related to occurrence and distribution of DWM in the upper Delaware River. Large lateral hydraulic gradients toward the river indicate the potential for substantial groundwater discharge at all three sites. Seeps and springs were present at or just upriver of the M reaches but not at the N reaches. Upward seepage through the riverbed measured with seepage meters was faster and more consistently upward at reaches populated by mussels. Median vertical hydraulic gradients were three to nine times larger at M reaches than at N reaches. Seepage determined two ways based on vertical temperature profiles at M reaches was circumneutral or predominantly upward. Riverbed temperature was generally colder in the M reaches than in the N reaches, particularly in discrete patches that were captured with the bed temperature snapshots. Combined, these results provide compelling evidence that groundwater discharge is substantial in areas populated



by DWM, but it is not evenly or universally distributed across the *M* areas. A paired seepage meter and streambed piezometer was installed within 1 m of an individual DWM at 3 specific locations, without evidence of anomalously strong seepage. Overall, the data indicate that DWM do not require focused groundwater discharge at their specific location, but

instead rely on the existence of substantial groundwater discharge within or just upstream of their populated area.

Other studies that have investigated the effect of groundwater discharge on benthic invertebrates have yielded mixed results. Some studies indicate a direct correlation between rate of groundwater discharge and abundance and taxonomic richness (Hunt et al., 2006), while others show little correlation (Schmidt et al., 2007). Few studies have related groundwater discharge with mussel abundance and species richness. A study conducted in a river with similarly coarse sediment indicated

a relation between mussel population density and upward seepage rate (Klos et al., 2015), but upward seepage in that setting was primarily driven by hyporheic exchange. Upward seepage at DWM sites in the Delaware River is primarily the result of groundwater discharge as evidenced by substantially colder water along M reaches relative to N reaches.

Direct measurements of groundwater discharge are difficult in settings such as the upper Delaware River where large boulders 1 m or more in diameter are common. Distinguishing hyporheic exchange from groundwater discharge is

particularly challenging (e.g., González-Pinzón et al., 2015; Menció et al., 2014; Ward et al., 2013; Bhaskar et al., 2012), hence the multiple lines of evidence pursued for this study. Therefore, few studies of exchange between groundwater and surface water have been successfully conducted in such coarse-grained sediments. Compared to those that have (e.g., Rosenberry et al., 2012; Fritz et al., 2009; Klos et al., 2015), values for point measurements of seepage exchange at these three sites on the Delaware River were not particularly large. This indicates that hyporheic exchange is perhaps smaller than

would be expected along M reaches, given the coarseness of the bed. And, just as was inferred regarding smaller-than-expected $K_v$ values, large horizontal hydraulic gradients adjacent to the river at all three sites would indicate larger amounts of groundwater discharge to the river, implying that substantially finer-grained sediment is present beneath the bed surface and is limiting flow. Fine- to medium-grained silt was attached to many of the M-site piezometers upon removal, but no silt was observed on removal of any of the N-reach piezometers. Silts are generally more electrically conductive than gravel and

cobbles, but the EMI data showed M zones were located over a mix of sediment types. The depth-integrated (~0-12 m) data presented here may not capture a shallow layer of fines. The multi-frequency GEM2 tool can be used at higher frequencies





for shallow depth-specific investigation (Briggs et al., 2016 (in press)), but the use of this higher-frequency data was complicated by the variable depth of surface water, which strongly influences the signals.

Cold-water anomalies were detected along all M reaches, but never along an N reach. At Site 2, mussel-location data from 2010 and 2012, in particular, indicated a strong clustering of animals directly adjacent to and downstream of the main spring described here and by Briggs et al. (2013) (Jeffrey Cole, unpublished data). DWM indeed may be present in these areas due to relatively stable and cold groundwater discharge that serves as a refuge for these animals during periods of lowest river stage. Additionally, mussel surveys have only been done at these locations during summer months; groundwater discharge also may offer benefits for mussel survival during cold winter extremes that are not apparent based on these data collected during the summer.

Data indicating flow in opposite directions across the riverbed are initially puzzling (Table 2). Vertical hydraulic gradients at in-river piezometers were small (and difficult to measure), as were most of the seepage rates measured with seepage meters. If riverbed sediment is substantially heterogeneous and highly transmissive, a likely situation in a cobble-bed river, it is not surprising that, at some locations, seepage meters indicated upward flow while temperature-profile data indicated downward flow (Rosenberry and Pitlick, 2009; Rosenberry et al., 2012; Angermann et al., 2012; Käser et al., 2009). This condition is not uncommon and indicates that exchange at those locations was largely driven by hyporheic processes, which is often superimposed on larger groundwater discharge patterns (Rosenberry et al., 2012). Hyporheic flow appeared to dominate exchange at the Site 3 N reach. Furthermore, substantial changes in the vertical component of hyporheic flow were indicated at most of the locations where temperature was measured at multiple depths in the riverbed (Table 2), also indicative of hyporheic exchange that is reduced or transitions to horizontal flow with increasing sediment depth (e.g., Briggs et al., 2012b).

## 5.1 Seepage at study sites relative to kilometer-scale values

Substantial groundwater discharge clearly occurs at areas populated by DWM, and no areas of focused discharge were identified immediately upstream or downstream of these three DWM-populated areas. However, is this prodigious discharge greater than what is typical along the upper reaches of the Delaware River? Fortunately, river discharge can be compared between two USGS gaging stations: Lordville (USGS station number 01427207; http://wterdata.usgs.gov/nwis) and



Callicoon. Several streams enter the river between these two gaging stations, most notably Little Equinunk and Basket

Creeks, but these streams are largely dry following prolonged dry periods. During 2013, the average gain in measured river

flow between Lordville and the downstream Callicoon stations was 5.9 m$^3$ s$^{-1}$. During October, the month with the smallest

average river flow, the gain between the two gaging stations was only 2.1 m$^3$ s$^{-1}$. Divided by the 29-km distance between the

two gaging stations, and assuming that stream inputs during this low-flow period were minimal, perhaps contributing half of

the increase in river discharge, this equates to an average increase in discharge of 2.2 L min$^{-1}$ per m of river reach. Assuming

that discharge of groundwater occurs equally on both sides of the river, this equates to an average rate of groundwater

discharge of 1.1 L min$^{-1}$ per m of river along each bank. This value is substantially smaller than most of the point-discharge

values that were measured in the various seeps and springs identified along each of these study sites where DWM have been

found (Table 3). Furthermore, only the seeps at and slightly above the river bank were identified either visually or with FLIR

data. Based on colder locations measured along M reaches with the profiler data, it is likely that other seeps also were

present but they were not observed because they were submerged. Therefore, although groundwater does contribute water to

other reaches of the upper Delaware River, the rate of discharge is substantially greater within areas populated by DWM.

The link between groundwater discharge and DWM preference for these areas remains unknown, warranting further research.

A recent study by Galbraith et al. (2015) indicates that DWM may be less mobile during dewatering caused by reduced river

flow than other mussel species. Groundwater discharge may offset the effects of dewatering of the riverbed caused by rapid

decreases in river stage.

## 6   CONCLUSIONS

1. *Alasmidonta heterodon* (dwarf wedgemussels) were located within or slightly downriver from reaches where a

prodigious amount of groundwater discharge was observed. Discrete, anomalously cold riverbed areas were detected in all

DWM reaches, but never in the reaches where DWM were not detected.    Measured discharges from individual seeps and

springs ranged from 0.5 to 77 L min$^{-1}$. Discharge from numerous other visible seeps was not measurable because it occurred

right at the bank or in river water that was less than 5 cm deep.

2. Horizontal hydraulic gradients measured at water-table wells installed within 12 m of the river were large and

indicated flow from groundwater to the river at all three study sites. Although gradient at Site 1 was measured only once,





gradients indicating flow toward the river at Sites 2 and 3 persisted year-round except for brief periods when they reversed in

response to abrupt river-stage rise following large rains or snowmelt.

3. Measurements of groundwater-surface-water exchange at specific points on the riverbed indicated that seepage was

upward across the sediment-water interface at 80 percent of DWM locations and 60 percent of non-DWM locations. Median

values of seepage along DWM reaches were 0.4 and 3.5 cm d$^{-1}$; median values of seepage at non-DWM reaches were 0.2 and

-0.01 cm d$^{-1}$. Vertical hydraulic gradients indicated upward flow at all locations in DWM reaches and median values were 3

to 9 times larger than at non-DWM reaches. Large rates of hyporheic exchange in places complicated the distinction between

groundwater discharge at DWM versus non-DWM reaches.

4. Discrete, anomalously cold riverbed areas were detected in all DWM reaches, but never in the reaches where DWM

were not detected. Streambed temperature-based seepage measurements guided by thermal surveys (e.g., at cold zones)

consistently indicated moderate groundwater upwelling to the river, confirming these as zones of rapid seepage.

5. Geology beneath the riverbed, as evaluated by bulk electrical conductivity, was variable at all three study sites, but

geologic variability did not appear to be correlated with distribution of DWM.

In conclusion, the collective lines of evidence indicate that DWM are situated in or directly downstream of areas of

substantial groundwater discharge to the river. The work presented here and in Briggs et al. (2013) may be the first to

demonstrate the importance of groundwater discharge to unionid species. Additional work is needed to better understand the

linkages between groundwater discharge and presence of DWM as well as geological controls that focus groundwater

discharge in these areas.

**Data availability**

Data on geomorphic parameters and groundwater-surface-water exchange are available upon request to Donald Rosenberry.

Temperature data, seepage rates determined from measurements of temperature, and geophysical data are available upon

request to Martin Briggs.

**Acknowledgements**



We thank Jeffrey Cole for advice and instructions related to river and riverbed logistics, and Heather Galbraith and

Carrie Blakesley for mussel identification and location; all from the USGS Northern Appalachian Research Branch. Don

Hamilton from the National Park Service Upper Delaware Recreational and Scenic River, and Joseph Markos, Richfield,

MN, are thanked for their field assistance and logistical support. Jason Halm's (University of Colorado- Boulder)

exceptional support before, during, and following field work is greatly appreciated. This work was funded by the U.S. Fish

and Wildlife Service. Use of trade names is for identification purposes only and does not constitute endorsement by the U.S.

Geological Survey.

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





**Tables**

Table 1 – Median values for parameters measured at each installation location at Sites 2 and 3.

| Site | Location | $u$ (m s$^{-1}$) | $h$ (m) | $q$ (cm d$^{-1}$) | $l$ (m) | $i$ | $K_v$ (m d$^{-1}$) | Med $u$ | Med $K_v$ | Med $h$ | Med $q$ | Med $i$ | $\tau$ (N m$^{-2}$) | $D_{50}$ (mm) | $\tau^*$ |
|---|---|---|---|---|---|---|---|---|---|---|---|---|---|---|---|
| | M1 | 0.18 | 0.52 | 0.43 | 0.28 | 0.0088 | 0.5 | | | | | | | | |
| | M2 | | 0.61 | -2.32 | | | | | | | | | | | |
| Site 2 | M3 | 0.17 | 0.57 | 0.52 | 0.44 | 0.0079 | 0.7 | 0.18 | 0.5 | 0.58 | 0.43 | 0.009 | 1.85 | 33 | 0.003 |
| | M4 | 0.23 | 0.58 | 0.13 | 0.18 | 0.0114 | 0.1 | | | | | | | | |
| | M5 | | 0.61 | 0.43 | | | | | | | | | | | |
| | N1 | 0.16 | 0.59 | 0.42 | 0.65 | 0.0031 | 1.3 | | | | | | | | |
| | N2 | | 0.61 | 0.14 | 1.15 | 0.0035 | 0.4 | | | | | | | | |
| Site 2 | N3 | 0.17 | 0.63 | 18.06 | 0.68 | 0.0044 | 40.9 | 0.17 | 1.3 | 0.59 | 0.18 | 0.003 | 1.70 | 4.4 | 0.024 |
| | N4 | | 0.57 | -0.38 | 1.09 | -0.0004 | 9.6 | | | | | | | | |
| | N5 | 0.17 | 0.50 | 0.18 | 0.36 | 0.0084 | 0.2 | | | | | | | | |
| Site 2 | S1 | 0 | 0.27 | 0.56 | 0.60 | 0.0168 | 0.3 | | | | | | | | |
| | S3 | 0.03 | 0.52 | 2.20 | | | | | | | | | | | |
| | M1 | 0.04 | 0.36 | 3.55 | 0.55 | 0.0009 | 39.4 | | | | | | | | |
| | M2 | 0.06 | 0.50 | 6.96 | 0.59 | 0.0118 | 5.9 | | | | | | | | |
| Site 3 | M3 | 0.08 | 0.41 | 3.5 | 0.42 | 0.0072 | 4.9 | 0.08 | 12.5 | 0.41 | 3.55 | 0.009 | 1.36 | 43 | 0.002 |
| | M4 | 0.11 | 0.38 | -0.12 | | | | | | | | | | | |
| | M5 | 0.19 | 0.50 | 26.92 | 1.39 | 0.0140 | 19.2 | | | | | | | | |
| | N1 | 0.26 | 0.38 | -0.01 | | | | | | | | | | | |
| | N2 | 0.34 | 0.46 | 84.22 | 0.56 | 0.0027 | 313.0 | | | | | | | | |
| Site 3 | N3 | 0.31 | 0.44 | 17.91 | 0.56 | 0.0009 | 202.2 | 0.31 | 202.2 | 0.44 | -0.01 | 0.001 | 4.71 | 55 | 0.005 |
| | N4 | 0.31 | 0.43 | -4.75 | 0.55 | -0.0119 | 4.0 | | | | | | | | |
| | N5 | 0.30 | 0.58 | -0.01 | | | | | | | | | | | |

$u$, current velocity; $h$, water depth at measurement location; $q$, seepage flux; $l$, depth of in-river piezometer screen beneath riverbed; $i$, hydraulic gradient at each in-river piezometer; $K_v$, vertical hydraulic conductivity determined at each in-river piezometer; $t$, reach-average shear stress; $D_{50}$, median grain size of bed surface; $t^*$, reach-average Shields stress.





Table 2 – Comparison of time-averaged VFLUX seepage values and median seepage-meter values at
select locations.

| Site | Location | Interval | VFLUX | Seepage meter |
|---|---|---|---|---|
| 2 | M3 | 0-10 | -69.2 | 0.5 |
| 2 | M3 | 10-20 | -12.1 | 0.5 |
| 2 | M3 | 20-40 | -9.9 | 0.5 |
| 2 | N3 | 0-10 | -32.2 | 18.1 |
| 2 | N3 | 10-20 | -18.6 | 18.1 |
| 2 | N3 | 20-30 | -10.2 | 18.1 |
| 2 | N3 | 30-50 | -5.0 | 18.1 |
| 3 | M2 | 0-10 | 4.4 | 7 |
| 3 | M5 | 10-20 | -11.1 | 26.9 |
| 3 | M5 | 20-30 | 0.5 | 26.9 |
| 3 | M5 | 30-50 | 32.5 | 26.9 |
| 3 | N3 | 0-20 | -19.7 | 17.9 |
| 3 | N3 | 20-30 | -6.9 | 17.9 |
| 3 | N3 | 30-50 | -4.4 | 17.9 |

Depth intervals are in cm and seepage rates in cm d$^{-1}$.

Table 3 – Rates of measured or calculated groundwater discharge. $\Delta Q$ is the difference in river flow
between two USGS gaging stations.

| Site | Measurement type | Measurement scale | Seepage rate*, L min$^{-1}$ |
|---|---|---|---|
| 1 | Spring | Point | 6.4 |
| 2 | Well | Site reach | 0.04 to 0.07 |
| 2 | Spring | Point | 12.9 |
| 2 | Spring | Point | 76.5 |
| 3 | Well | Site reach | 3E-5 to 6E-5 |
| 3 | Spring | Point | 0.5 to 2 |
| 1-3 | $\Delta Q$ | 29 km reach | 1.1 |

* Reach-scale seepage rate is per meter of river reach per single side of river



**Figure captions**

Fig. 1 – Delaware River reach (highlighted) on the border between New York and Pennsylvania between

Hancock and Callicoon.

Fig. 2 – Sites 1 (panel A), 2 (panel B) and 3 (panel C). Arrows indicate direction of river flow.

Fig. 3 – Median daily river discharge ($Q$) based on the period of record 1975-2012 (USGS station number

01427510; http://waterdata.usgs.gov/nwis). Daily-average $Q$ also is plotted during June-July 2012 and

June-July 2013. Periods of site visits during June 22-July 1, 2012, July 20-30, 2012, and June 28-July 2,

2013, are shown as blue rectangles.

Fig. 4 – EMI quadrature data at 33,030 Hz converted to bulk conductivity for (A) Site 1, (B) Site 2, and

(C) Site 3. Warmer colors indicate less conductive streambed material potentially correlating to coarse-

grained surficial deposits and bedrock.

Fig. 5 – (A) Photograph of riverbank at Site 2 with red rectangle indicating area of infrared image, (B),

Color infrared image with blue area showing colder groundwater entering the river. Color scale indicates

temperature, in °C.

Fig. 6 – River stage, water-table elevation, and hydraulic gradient at Sites 2 and 3. Legend in panel B

also applies to panels A and C except Site 3 data are from the WT2 monitoring well. (A) Site 2, July

2012 through June 2013; (B) 20-minute data from Site 2 showing gradient reversal on September 19,

2012; (C) Site 3, July 2012 through June 2013.

Fig. 7 – Median values of seepage flux. Error bars indicate maximum and minimum measured values.

Median value for Site 3 N2 is 84 cm d$^{-1}$.

Fig. 8 – Seepage rates determined with VFLUX and Delaware River discharge determined over 25-day

period from June 28-July 23.

Fig. 9 – Riverbed temperatures indicated by snapshot thermal surveys (shaded riverbed areas) and FO-

DTS at Site 2 (panels A and B) and Site 3 (panels C and D). Colored circles in panels A and C indicate

temperature and sizes of circles in panels B and D indicate temperature standard deviation during 4- and

2-day cable deployments at Site 2 and 3, respectively.



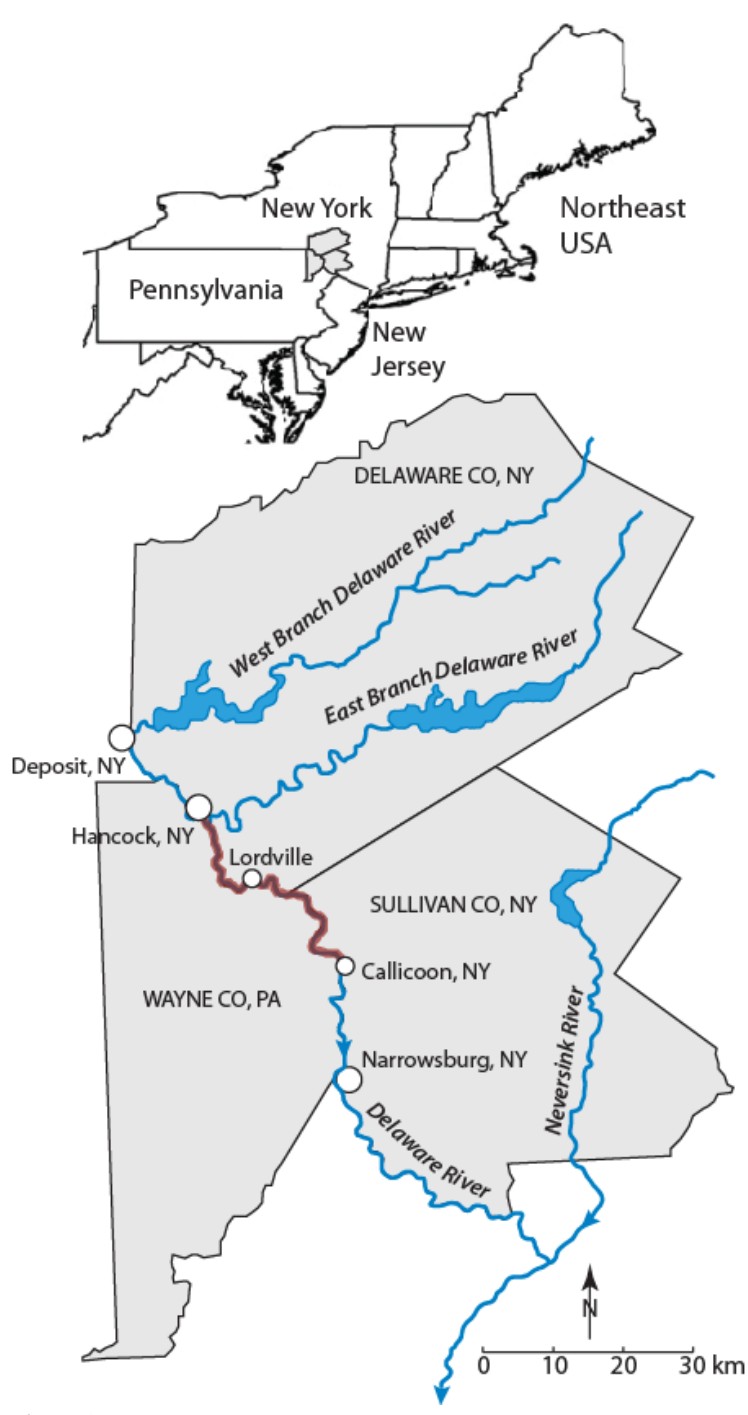

Figure 1



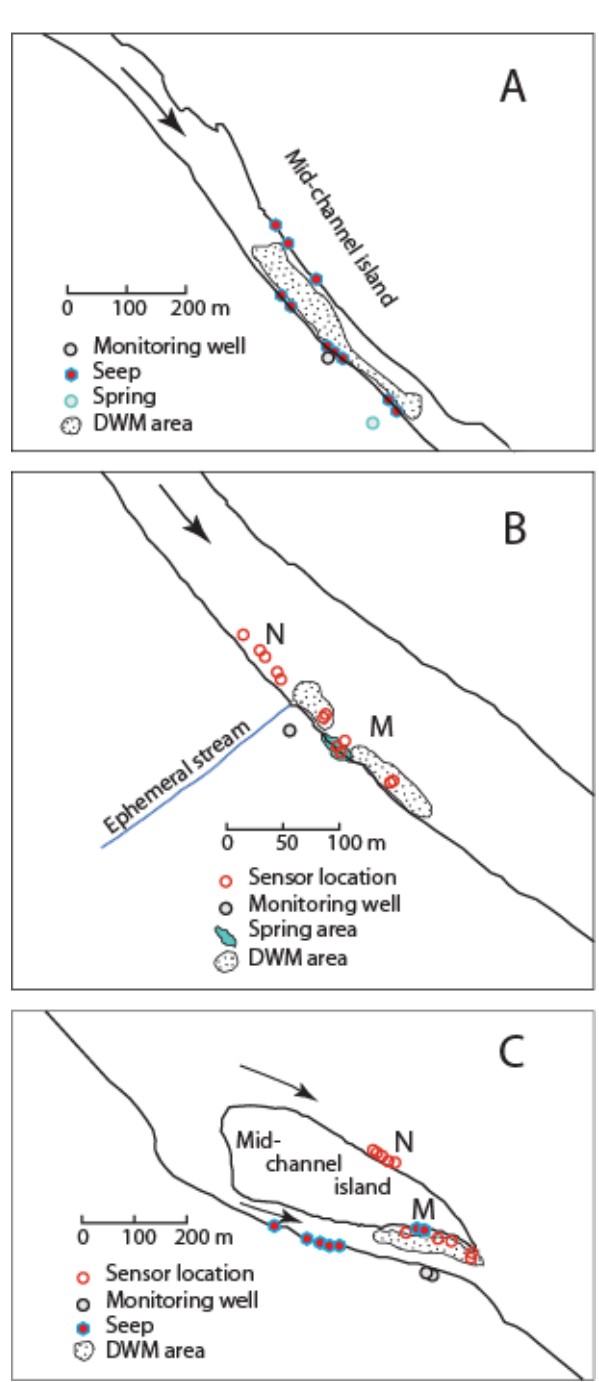

Figure 2





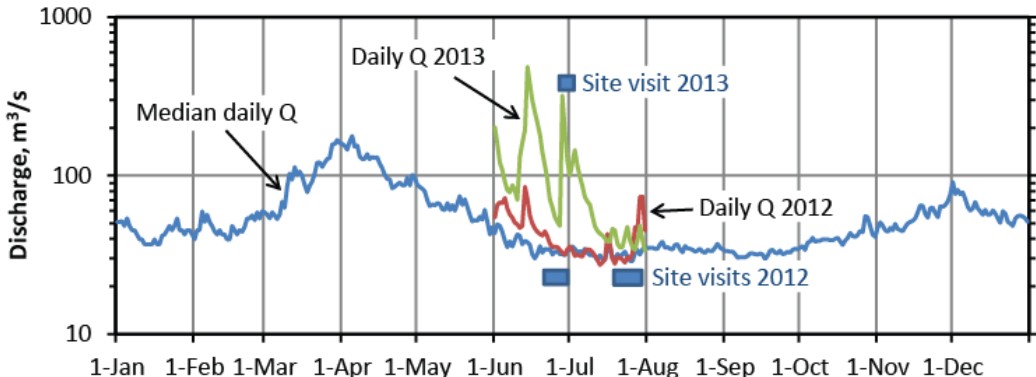

Figure 3

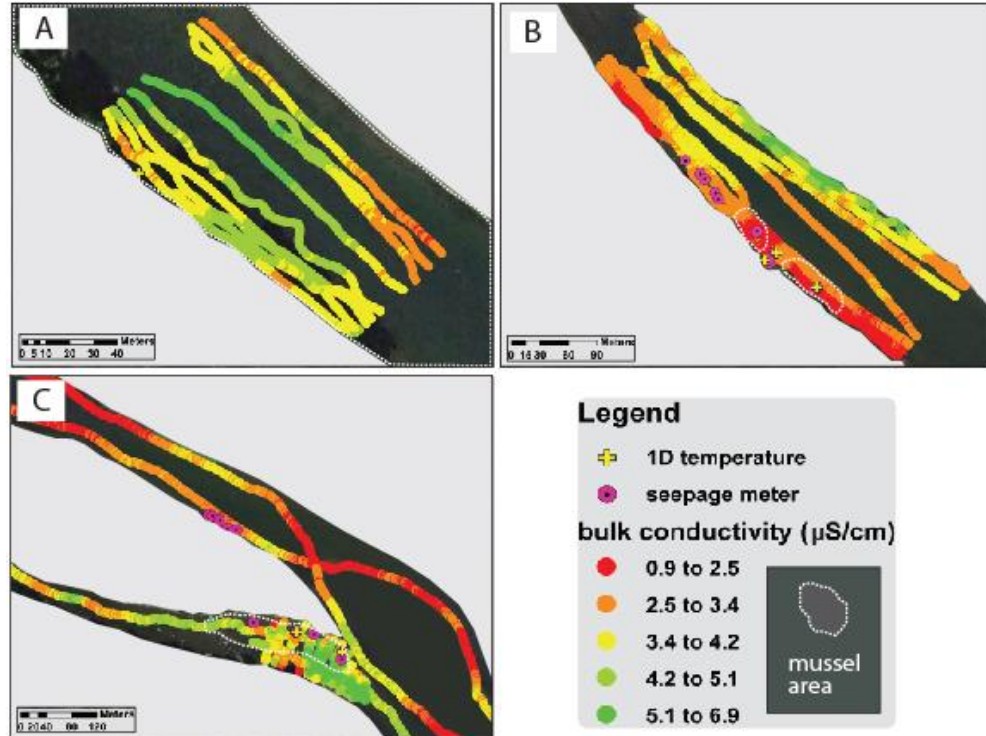

Figure 4





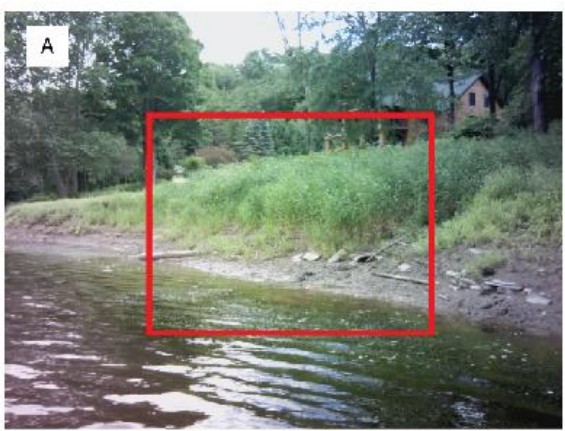

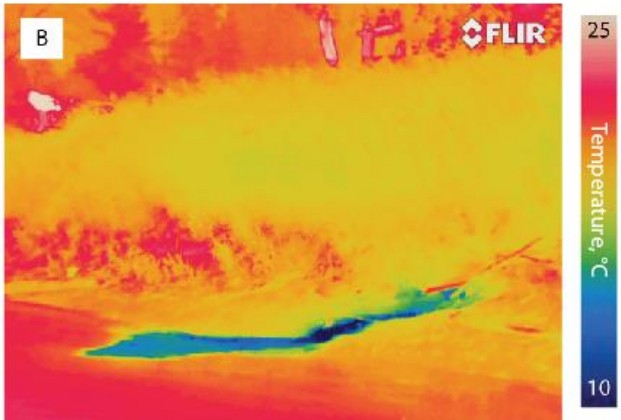

Figure 5





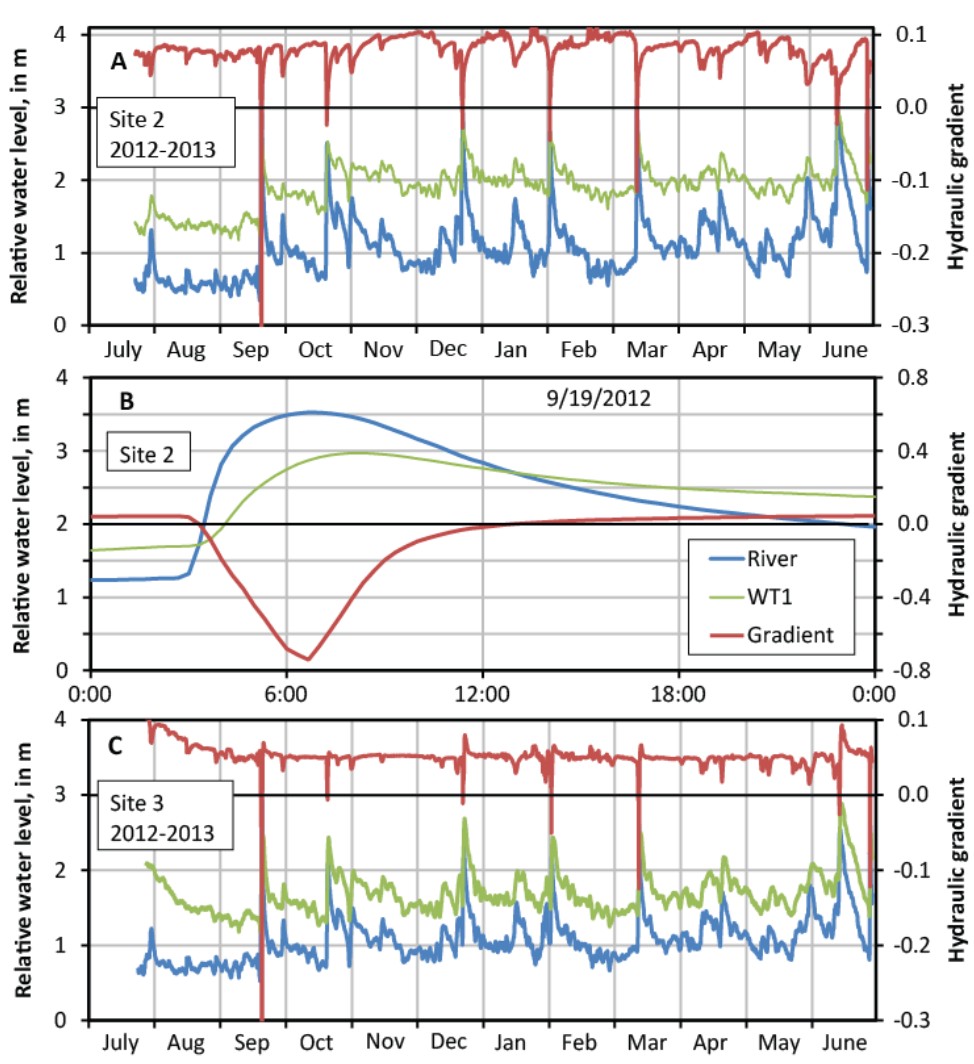

Figure 6



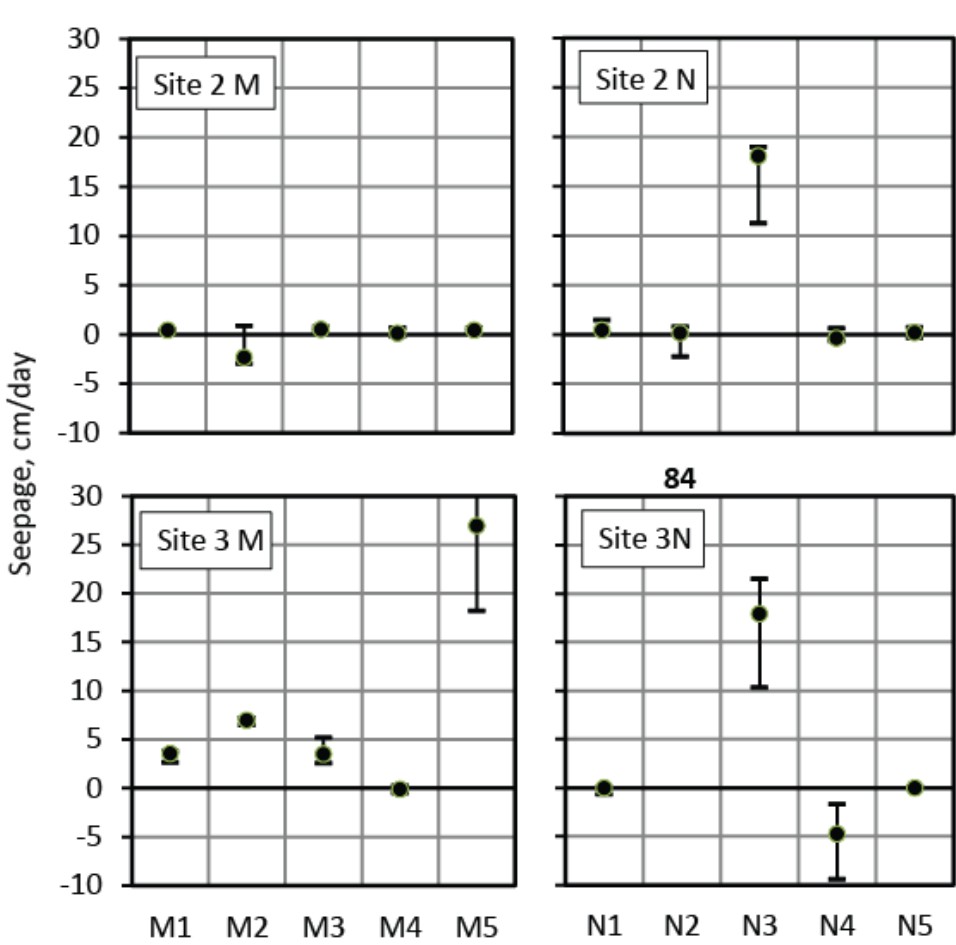

Figure 7





765

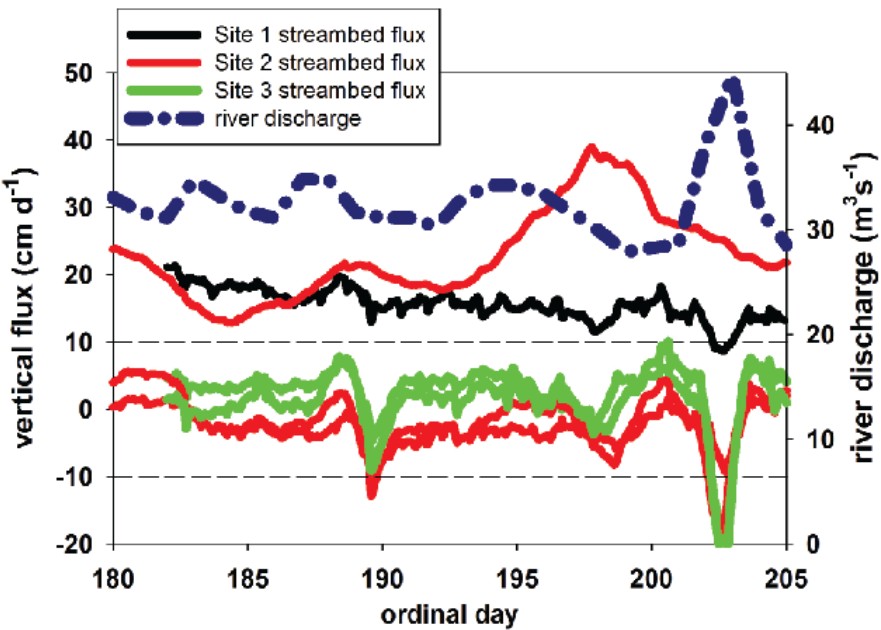

766
767     Figure 8
768



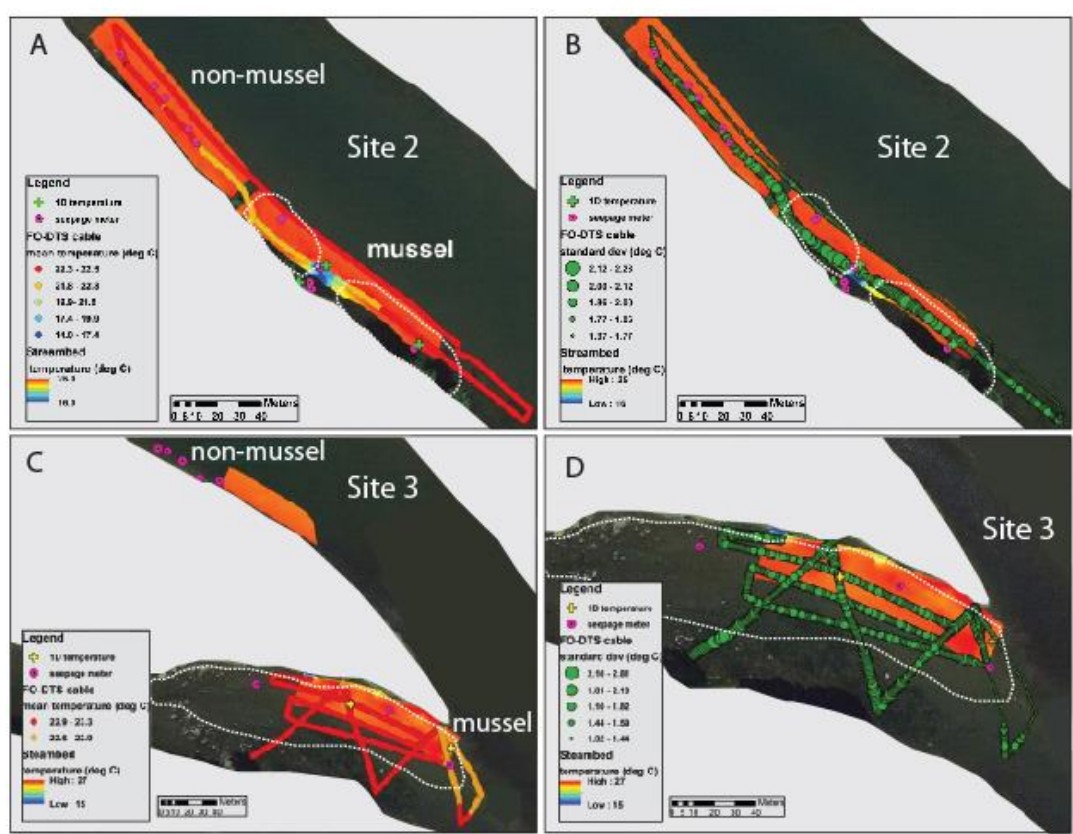

769
770      Figure 9