# Peer review of "Influence of groundwater on distribution of dwarf wedgemussels $(Alasmidonta\ heterodon\ )$ in the upper reaches of the Delaware River, northeastern USA"

_Hydrology and Earth System Sciences, 2016_

## Referee Comment (RC1) · Anonymous Referee #1 · 31 May 2016

General Comments

The authors used a variety of techniques to measure groundwater discharge at locations where dwarf wedgemussels (DWM) are known to be present or not present. While there is natural uncertainty in several of the measurements made, and in some cases conflicting results, generally the DWM are located where cold groundwater discharges. Overall, the paper is well written, but the supporting figures require some attention to bring the paper up to publication standards.

I must admit that I was somewhat underwhelmed with the paper. While numerous

results were reported, and shown to be generally supportive of the hypothesis that DWM are located in areas of greater-than-normal groundwater discharge, I was hoping for some interesting comparison of the measurements. It wasn't until the conclusions that such a comparison was made. While the broad range of techniques adds value to the paper, I find it somewhat narrow in scope, and essentially a case study.

Specific Comments

Figure 2 – the text (lines 172-175) refers to locations were seepage meters were installed. However, the figure itself only mentions seeps, which I take to mean actual locations where seepage was observed. Moreover, on line 175, the authors mention locations S1 and S2, but no such locations are shown on the figure.

Line 181 – the opening paragraph indicates that pressure is measured in streambed piezometers. Actually, this is incorrect. Head is measured. The authors say this later in the paragraph (line 187) – vertical head differences.

Figure 4 – The locations of the seepage meters and 1D temperature sensors cannot be seen on this figure. Some other symbol color (blue?) is needed to differentiate from the conductivity. However, I don't think that the instrument locations should be shown in this figure anyway – see my next comment. The EMI results are quite separate from the thermal results and are not even discussed together.

Figure 2 should be modified or a new figure included to show the locations of measurements of seepage (numbered according to Table 1). Note, locations M4 and M5 are also referred to in the text (line 260) but are not shown anywhere. This figure could also show the 1D sensor locations.

Line 314-315, and Line 323-324 – stronger justification is needed about the cross-sectional area. Why 10 to 20 m. Where is the evidence of this? I realize that a range is given, but thickness seems like a rather important parameter.

Technical corrections

Line 53 – low flows – there is no need to hyphenate here.

Line 79 – 100s

Line 94 – relate groundwater – surface-water

Line 107 (Fig. 2A). Bracket missing.

Line 110 – (Fig. 2B). left bracket missing.

Line 125 – lower case for header

Line 135 – two furthers.

Line 143 – data were

Line 160 – some explanation of how A was assessed is needed. How large was the vertical plane? Given that the volumetric seepage rate is being compared between sites, it might be better to compare the flux.

Line 179 – lower case for header

Line 218 – (July 25 to July 27, 2012)

Line 243 – perhaps put (or magnetic susceptibility) in brackets or don't mention it at all, as this property was not measured in this study.

Line 267 – why weren't the river slope and water depths measured at Site 1?

Line 280 Groundwater – surface-water exchange

Line 275 – authors should refer to Fig 2 which shows the seep locations.

Line 323 – should spell out three.

Line 529 – groundwater – surface-water

Line 357 – iv should be defined earlier in line 191.

Line 389 – four not 4

Line 392 – four days

Line 399 – two days

Line 433 – three specific locations

Line 503-504. This sentence is repeated again in point 4 of the conclusions.

Table 1 – what are all the green triangles in the table. This looks like an excel table that has been cut and pasted into the text.

Table 2 – include the units in the table headers rather than at the bottom as this is inconsistent with the other tables.

Figure 7 – units should be cm d-1 for consistency with unit notation used elsewhere.

Figure 8 – the line width is much larger for this figure than the other figures. It should be reduced.

---

## Referee Comment (RC2) · Anonymous Referee #2 · 21 Jun 2016

General comments: The paper reports an interesting study on the correlation between mussel occurrence and groundwater discharge in and to a large river.

The methods used to study and quantify the river-groundwater interactions are not new, but one of the main points in the paper is that it clearly demonstrates the need for applying several different methods to map and estimate seepage patterns and rates.

I am not entirely convinced about the general conclusions that there is compelling evidence that diffuse groundwater discharge is responsible for the occurrence of mussel habitats in the river. Or, the argumentation is not exactly clear. From my understand-

ing it is exactly not the diffuse discharge but seeps or areas with focused groundwater discharge that sustain these mussel populations. The data is not always pointing to the same observation, so this needs more clarification and is why I recommend major revision, because the authors may not agree to this.

It was also interesting to see that simple manual temperature readings of the river bed actually provided a good mapping of the major inflow zones. We tend to use advanced methods like DTS or UVAs to do this, but keep it simple seems to work nicely – and maybe cheaper and more effective?

Specific comments: L133: The piezometers where groundwater samples were collected are not shown on Figure 2? L141: Suspended sediment. That suggests to me a very high discharge. Were they found at all three sites? L175: Are these places with suspended sediments? L310-318: You estimate a rather low K value (0.063 m/d), which I think represents silt more than sand. You call it different things; sandy sediments, silty sand, sand and fractured shallow sandstone, in this section. Maybe it is not so important what it is called, more so that this diffuse discharge is low? At least, here I am left with the impression that it is not the diffuse groundwater discharge that sustains DWM populations? L319-327: The same goes for site 3 with even lower discharge. Maybe the last sentence is important in this context and for the whole study. That it is not slow, diffuse groundwater discharge that is responsible for sustaining DWM populations, but, rather the preferential flow paths. L337-339: Unclear. L344-345: These "median" values are also the first and last entries in Table 1 – is that just a coincidence? L345: I find that there are other sites with "consistently" upward seepage, so does consistently mainly refer to "substantial", i.e., they show high fluxes and does not refer to "direction"? L367: Is it fair to say that this indicates a greater potential for seepage at N locations, which goes against your conclusion? L374: With my experience some of the fluxes are not just moderate, but also high. L382: How is this error bound explained? Maybe you should also put an error bound on the seepage meters? L413: An interesting observation. L423-425: I am not entirely convinced about this

conclusion. To me your data and observations collectively suggest that it is not the diffuse groundwater discharge (rather low), but known seeps and unknown preferential flow paths that are related to DWM occurrence. (1) Seeps, I agree. Your observations support this. (2) fluxes measured by seepage meters. I am not quite convinced as you measure positive/negative fluxes in all places and by the fact that they do not compare well with fluxes estimated from temperature profiling. (3) Hydraulic gradients. Maybe not the best measure as the flux will depend on Kv. And, Kv was higher at N sites? L434-435: Which brings me to this conclusion, which I am not sure I fully understand or agree with. On the one hand, you argue that DWM rely on "substantial" discharge, which I read as high fluxes; on the other hand, that DWM do not rely on focused (=high?) discharge. L441-442: Upward seepage .. is primarily the result of groundwater discharge. Is that not obvious, or are you referring to the possibility of hyporheic flow? L461-462: Exactly here you argue that a strong clustering of animals is related to the occurrence of springs, I can agree with this. L468-470: Why will the methods produce two different results because of a cobble-bed river? L471: Now you argue that hyporheic flow can dominate, see comments above? L490: Are you then saying that discharge cannot happen uniformly/diffusively, but must occur as springs, focused flows, through preferential flow paths?

Technical corrections: L153 and Figure 2: Should the figure legend say "hole" instead of monitoring well? L350: Maybe say larger instead of faster (like in the sentence just below)? L358: Maybe help the reader by saying "q estimated from seepage meters .." L379 and Figure 8: There are three red curves in the figure, but only one legend?

---

## Author Comment (AC1) · 9 Aug 2016

We thank the reviewers for their thorough and thoughtful comments.

Responses to Anonymous Referee comments on "Influence of groundwater on distribution of dwarf wedgemussels (Alasmidonta heterodon) in the upper reaches of the Delaware River, northeastern USA"

Responses to Anonymous Reviewer #1 general comments

Reviewer 1 indicated being underwhelmed by the manuscript due to the lack of con-

cordance among different types of data and the lack of comparison of types of results. Regarding lack of concordance, that perhaps should not be unexpected. We clearly did not properly indicate the degree of difficulty in not only measuring fluxes in a high-energy, coarse-grained fluvial environment, but in separating flows across the sediment-water interface due to hyporheic exchange from flows generated by larger-scale hydraulic gradients within the aquifer. We will add text to indicate that this difficult challenge is rarely met in these settings and that it requires multiple lines of evidence to reach reliable and defensible conclusions regarding the degree to which groundwater discharge is associated with presence of this endangered species. As to lack of comparison of results, we agree with the reviewer. We should have compared results to a greater extent and are glad to do so, indicating how seepage meters quantify flux at the interface, no matter the cause of that flux, and that temperature of the bed is a good indicator of groundwater discharge so long as hyporheic exchange does not overwhelm the groundwater-discharge signal. We will also indicate more clearly that we were expecting that porewater chemistry would be a great tool to distinguish sources of water but, alas, there was virtually no difference in chemistry of the various water types in this system.

As to the comment that the manuscript is narrow in scope and essentially a case study, we respectfully disagree. A case study is one that takes a concept or method that has been developed previously and applies that to a new location. Our study is novel in that it applies a broad range of techniques for quantifying exchange between groundwater and surface water to a fluvial environment where such measurements are very challenging to obtain. Furthermore, this multi-method approach was applied to a pressing hydroecological question regarding the potential tie between groundwater discharge and patchy occurrence of the endangered dwarf wedgemussel species. The range of results and the sometimes lack of agreement of results are further evidence and a good example of the difficulty associated with conducting research in such a setting. One major conclusion is that all of these methods are necessary to reach comprehensive and defendable conclusions regarding groundwater discharge in relation to mussel

beds. Lastly, as to the lack of inter-method comparison and discussion of results, the reviewer is correct. We will provide a much more thorough discussion of methods comparisons, including reasons for lack of concordance in some cases, in the discussion section.

Responses to specific Reviewer #1 comments

Reviewer 1: Figure 2 – the text (lines 172-175) refers to locations were seepage meters were installed. However, the figure itself only mentions seeps, which I take to mean actual locations where seepage was observed. Moreover, on line 175, the authors mention locations S1 and S2, but no such locations are shown on the figure.

Authors' reply: The legends in Figure 2 present symbols that pertain to locations where sensors were installed. We can add text to the figure caption to make that more clear. We can also provide background shading for the two symbols situated within the spring area in panel B to make them more visible. Regarding labeling of specific sensor locations, we indicated our naming procedure in the text as there is not enough room in the figures to add that level of detail. However, we can add text to indicate that locations S1 and S3 also increase in number with distance downstream. Or, if the journal allows, we can substantially increase the size of the figure panels, in which case we will be able to add labels for specific sensor locations.

Reviewer 1: Line 181 – the opening paragraph indicates that pressure is measured in streambed piezometers. Actually, this is incorrect. Head is measured. The authors say this later in the paragraph (line 187) – vertical head differences.

Authors' reply: The reviewer is correct. Although the sensor actually measures pressure in kPa, we convert that value to hydraulic head and should have described the measurement as such.

Reviewer 1: Figure 4 – The locations of the seepage meters and 1D temperature sensors cannot be seen on this figure. Some other symbol color (blue?) is needed to

differentiate from the conductivity. However, I don't think that the instrument locations should be shown in this figure anyway – see my next comment. The EMI results are quite separate from the thermal results and are not even discussed together.

Authors' reply: We can easily change the symbols for seepage meters and temperature profilers to make them more visible. We had not realized the extent to which the symbols become lost when reduced to page size. As to whether to include the symbols on the figure, we prefer to include the locations of these sensors. They provide a useful indication of the relative positioning of measurements of seepage discharge with an indication of underlying geology that may be related to seepage discharge.

Reviewer 1: Figure 2 should be modified or a new figure included to show the locations of measurements of seepage (numbered according to Table 1). Note, locations M4 and M5 are also referred to in the text (line 260) but are not shown anywhere. This figure could also show the 1D sensor locations.

Authors' reply: We explain on lines 173-174 that measurement locations are numbered 1 through 5 with numbers increasing with distance downstream. We could add labels in Figure 2 to indicate specific measurement locations but we think the figure would become too cluttered. Given that there are 5 circles at each M and N site, it is reasonably easy to determine which circle in the figure pertains to a specific sensor location.

Reviewer 1: Line 314-315, and Line 323-324 – stronger justification is needed about the crosssectional area. Why 10 to 20 m. Where is the evidence of this? I realize that a range is given, but thickness seems like a rather important parameter.

Authors' reply: We agree that thickness is an important parameter, but there is not much information related to the thickness of the surficial soils plus the portion of the underlying bedrock that is fractured and transmits water. Based on observations of outcrops and soil surveys of the area, our best estimate of the vertical section that readily transmits water is 10 to 20 m. We can add a citation to a report titled "Soil survey of Wayne County Pennsylvania from which we obtained information about thickness of

glacially deposited sediments above the bedrock.

Reviewer #1 Technical corrections

Authors' reply: We thank the reviewer for making editorial corrections to the manuscript and we are happy to accept all editorial suggestions and revise accordingly.

Responses to Anonymous Reviewer #2 general comments

Anonymous reviewer #2 also was concerned about the lack of clarity regarding all datasets pointing to the conclusion that groundwater discharge is correlated with presence of endangered mussels. This further compels us to better and more clearly explain that it took the full suite of methods and lines of evidence to reach this conclusion. We can also more emphatically indicate that mussels are present in areas of rapid and/or focused groundwater discharge, in addition to diffuse groundwater discharge. We also were pleased by the success of simply making manual temperature measurements of the riverbed and yet we failed to compare results from manual bed-temperature surveys compared with the much more expensive and sophisticated fiber-optic measurements. We will point out this comparison and include likely reasons for the differences in results.

Responses to specific Reviewer #2 comments

Reviewer 2: L133: The piezometers where groundwater samples were collected are not shown on Figure 2?

Authors' reply: Locations of piezometers are indicated in Figure 2 but are not individually labeled due to the scale of the figure. We have added a sentence to the figure caption to better explain this: "Sensor locations pertain to seepage-meter and in-river piezometer installations."

Reviewer 2: L141: Suspended sediment. That suggests to me a very high discharge. Were they found at all three sites?

Authors' reply: Indeed it does. Suspended sediment, sometimes described as "boiling sand", is indicative of a seepage rate large enough that sediments are lifted off of the bed. This condition was observed at Sites 1 and 3. At Site 2, the rapid groundwater discharge occurred just above the river surface and visibly flowed downslope to the river. At all sites, zones of strong groundwater discharge were spatially discrete at the centimeter scale. We will point out the strong relevance of this important feature that provides strong evidence of focused groundwater discharge in these areas.

Reviewer 2: L175: Are these places with suspended sediments?

Authors' reply: Although the spring area had attributes of focused and rapid groundwater discharge, namely very cold water and soft sediments, we did not observe suspended sediment within the spring area. Although groundwater discharge was obviously prodigious, it was not rapid enough to suspend sediments at the submerged portion of the riverbed.

Reviewer 2: L310-318: You estimate a rather low K value (0.063 m/d), which I think represents silt more than sand. You call it different things; sandy sediments, silty sand, sand and fractured shallow sandstone, in this section. Maybe it is not so important what it is called, more so that this diffuse discharge is low? At least, here I am left with the impression that it is not the diffuse groundwater discharge that sustains DWM populations?

Authors' reply: The sand description was based on sediment removed during well installation. The silty sand description was based on the hydraulic-conductivity value from the slug test. The fractured sandstone was based on the observation that the surface of the sandstone bedrock typically is relatively more fractured. We had indicated that sediment was sandy during well installation and that bedrock was composed primarily of sandstone in an earlier version of the manuscript but we were clearly overzealous in paring the manuscript for submission to HESS. We can easily replace that information for clarity. However, the bottom line, as the reviewer surmised, is that

the sediments adjacent to the river are relatively fine, indicating that diffuse ground-water discharge like is not particularly large in these areas, suggesting that focused, more rapid discharge is more likely related to presence of this endangered species of mussel.

Reviewer 2: L319-327: The same goes for site 3 with even lower discharge. Maybe the last sentence is important in this context and for the whole study. That it is not slow, diffuse groundwater discharge that is responsible for sustaining DWM populations, but, rather the preferential flow paths.

Authors' reply: The reviewer is correct in assuming that diffuse GW discharge indeed likely is less influential in determining the presence/absence of these endangered mussels. We will do a better job of emphasizing that observation in a revised version.

Reviewer 2: L337-339: Unclear.

Authors' reply: We may have gotten a bit down in the weeds with our description of these flow reversals. Our main point was that they were slow-lived and that, even only 18 hours after peak river stage, when river stage was still 0.75 m higher than the pre-event stage, the normal upward hydraulic gradient had already become established. We can revise the text to better emphasize this point.

Reviewer 2: L344-345: These "median" values are also the first and last entries in Table 1 – is that just a coincidence?

Authors' reply: How interesting. That indeed is purely coincidental.

Reviewer 2: L345: I find that there are other sites with "consistently" upward seepage, so does consistently mainly refer to "substantial", i.e., they show high fluxes and does not refer to "direction"?

Authors' reply: A good point. We should have stated that this reach had seepage that was consistently upward at substantial rates. Other reaches had a preponderance of upward seepage values but many were small with values well less than 1 cm/d.

Seepage at all locations but one at the M reach of Site 3 was upward at rates of 3.5 to nearly 27 cm/d. We can revise to write "The only reach where seepage was substantially upward was the M reach at Site 3, where upward seepage at 4 of 5 locations was greater than 3 cm/d."

Reviewer 2: L367: Is it fair to say that this indicates a greater potential for seepage at N locations, which goes against your conclusion?

Authors' reply: We don't think so. Although median $K_v$ at the N reach of Site 2 was larger than at the M reach, both values were essentially the same (1.3 and 0.5), given the uncertainty associated with determining K. The much larger $K_v$ at the N reach at Site 3 indicates that water can flow more easily through those sediments, but the median vertical hydraulic gradient was essentially zero at the N reach, indicating a small potential for seepage.

Reviewer 2: L374: With my experience some of the fluxes are not just moderate, but also high.

Authors' reply: The largest value was just over 84 cm/d, which indeed is a large number for lake settings. However, numerous studies have reported seepage in other fluvial settings that is much faster, with values well over 100 cm/d. Given the coarseness of the bed and the relatively fast river current along the Site 3 N reach, we were actually expecting to measure faster seepage than we did in response to hyporheic exchange. That is why we chose to call seepage at that reach moderate rather than fast.

Reviewer 2: L382: How is this error bound explained? Maybe you should also put an error bound on the seepage meters?

Authors' reply: The temperature-based flux expected error range was determined for similar sediments using a Monte Carlo analysis where sediment thermal properties were varied simultaneously within expected ranges. This previous work (Briggs et al 2012b) will now be cited at the end of this sentence. Errors associated with seepagemeter measurements depend on the setting in which measurements are conducted. Figure 7 presents median values along with maximum and minimum values at each location. We think this figure nicely indicates the relative uncertainty associated with seepage-meter measurements at each location.

Reviewer 2: L423-425: I am not entirely convinced about this conclusion. To me your data and observations collectively suggest that it is not the diffuse groundwater discharge (rather low), but known seeps and unknown preferential flow paths that are related to DWM occurrence. (1) Seeps, I agree. Your observations support this. (2) fluxes measured by seepage meters. I am not quite convinced as you measure positive/negative fluxes in all places and by the fact that they do not compare well with fluxes estimated from temperature profiling. (3) Hydraulic gradients. Maybe not the best measure as the flux will depend on Kv. And, Kv was higher at N sites?

Authors' reply: We agree that observed seeps and springs are probably the strongest evidence that groundwater discharge is much greater in areas populated by endangered mussels. We also think seepage-meter data are fairly convincing. Seepage meter fluxes, although clearly influenced by hyporheic exchange, collectively indicate greater upward seepage at M reaches than at N reaches. Median seepage at the M reach at Site 2 was more than double the N reach value and median seepage at the M reach at Site 3 was the largest of all, while the median seepage at the corresponding N reach was slightly downward. The "noise" attributed to hyporheic exchange in other fluvial settings also has been dealt with in other studies by averaging values across sites to separate the larger-scale groundwater exchange from the hyporheic exchange. We agree that basing seepage rates on hydraulic gradients in these settings is suspect as the hydraulic gradients commonly are very small and difficult to measure. Seepage determined from vertical-temperature profiling also can be problematic due to non-vertical flow associated with hyporheic exchange. So it is not surprising that temperature-based seepage did not always compare with physically based seepage. We discuss these issues on lines 466-476.

Reviewer 2: L434-435: Which brings me to this conclusion, which I am not sure I fully understand or agree with. On the one hand, you argue that DWM rely on "substantial" discharge, which I read as high fluxes; on the other hand, that DWM do not rely on focused (=high?) discharge.

Authors' reply: The point of lines 434-435 is that individual animals do not seem to be located precisely at a point of focused groundwater discharge but instead rely on one or numerous points of focused groundwater discharge that occur nearby. We obviously need to revise the text to make this point more clearly, such as "These paired measurements indicate that individual DWM may not require focused groundwater discharge to occur precisely at their location, but instead may rely on the existence of focused groundwater discharge within or just upstream of their populated area."

Reviewer 2: L441-442: Upward seepage .. is primarily the result of groundwater discharge. Is that not obvious, or are you referring to the possibility of hyporheic flow?

Authors' reply: This text was comparing conditions at the Delaware River with seepage rates reported in another study. Taken out of context (or evidently even in context), the sentence is confusing. We can revise to write "The net upward seepage at DWM sites in the Delaware River, although clearly influenced by hyporheic exchange, is primarily the result of groundwater discharge as evidenced by substantially colder water along M reaches relative to N reaches."

Reviewer 2: L468-470: Why will the methods produce two different results because of a cobble-bed river?

Authors' reply: This is a very good question and one that is often difficult to understand. Seepage meters measure flux across the bed no matter the actual vector direction of the flow. Water can be flowing primarily horizontally beneath the bed and a seepage meter will measure whatever water crosses the sediment-water interface beneath the meter. However, hydraulic gradients and seepage based on vertical temperature profiling assumes that flow through the sediment is vertical. Furthermore, they integrate

the flow based on the positioning of the well screen or temperature sensors, which may be well beneath the bed and influenced by subsurface flowpaths that are not related to flow at the bed surface at that specific location. Others papers also have observed this feature that is a characteristic of some hyporheic settings. We clearly did not properly explain this situation. We propose the following revision that hopefully conveys this concept more clearly: "Data indicating flow in opposite directions across the riverbed are initially puzzling (Table 2). Some of the discordant data may be attributed to measurement error. Vertical hydraulic gradients at several in-river piezometers were very small (and difficult to measure), as were some of the seepage rates measured with seepage meters. Furthermore, hyporheic flowpaths in substantially heterogeneous and highly transmissive sediment, a common situation in a cobble-bed river, is predominantly horizontal with small upward and downward flow components. Because piezometers and vertical temperature profilers are installed vertically, interpretations of hydraulic gradient and seepage assume vertical flow through the sediments, often a poor assumption in hyporheic settings. It is not uncommon for seepage meters to indicate upward flow while hydraulic gradients indicate downward flow (Rosenberry and Pitlick, 2009; Rosenberry et al., 2012; Angermann et al., 2012; Käser et al., 2009). Locations with discordant data indicate flow across the sediment-water interface was largely driven by hyporheic processes, which is superimposed on larger-scale groundwater discharge patterns (Rosenberry et al., 2012)."

Reviewer 2: L471: Now you argue that hyporheic flow can dominate, see comments above?

Authors' reply: Yes, particularly at the N reach of Site 3 where cobbles and K values were largest.

Reviewer 2: L490: Are you then saying that discharge cannot happen uniformly/diffusively, but must occur as springs, focused flows, through preferential flow paths?

Authors' reply: Not quite. We are saying that hyporheic exchange greatly affects measurements and interpretations of larger-scale groundwater discharge to the river. Perhaps due to hyporheic exchange, or perhaps due to underlying geologic heterogeneity, focused groundwater discharge also occurs, likely in addition to diffuse groundwater discharge. It seems that DWM are located in areas where locations or amounts of focused groundwater discharge are particularly numerous or large.

Reviewer #2 Technical corrections

We agree with all of the technical/editorial suggested by Reviewer #2 and will readily make all suggested changes to text and figures.

Please also note the supplement to this comment:
http://www.hydrol-earth-syst-sci-discuss.net/hess-2016-187/hess-2016-187-AC1-supplement.pdf

––––––––––––––––––––

---

## Author Response (AR1)

Responses to Anonymous Referee comments on "Influence of groundwater on distribution of dwarf wedgemussels (*Alasmidonta heterodon*) in the upper reaches of the Delaware River, northeastern USA"

**Responses to Anonymous Reviewer #1 general comments**

Reviewer 1 indicated being underwhelmed by the manuscript due to the lack of concordance among different types of data and the lack of comparison of types of results.  Regarding lack of concordance, that perhaps should not be unexpected.  We clearly did not properly indicate the degree of difficulty in not only measuring fluxes in a high-energy, coarse-grained fluvial environment, but in separating flows across the sediment-water interface due to hyporheic exchange from flows generated by larger-scale hydraulic gradients within the aquifer.  We have added text to indicate that this difficult challenge is rarely met in these settings and that it requires multiple lines of evidence to reach reliable and defensible conclusions regarding the degree to which groundwater discharge is associated with presence of this endangered species.  As to lack of comparison of results, we agree with the reviewer.  We should have compared results to a greater extent and have now done so in a new section of the Discussion, indicating how seepage meters quantify flux at the interface, no matter the cause of that flux, and that temperature of the bed is a good indicator of groundwater discharge so long as hyporheic exchange does not overwhelm the groundwater-discharge signal.  We have also indicate more clearly that we were expecting porewater chemistry would be a great tool to distinguish sources of water but, alas, there was virtually no difference in chemistry of the various water types in this system.

As to the comment that the manuscript is narrow in scope and essentially a case study, we respectfully disagree.  A case study is one that takes a concept or method that has been developed previously and applies that to a new location.  Our study is novel in that it applies a broad range of techniques for quantifying exchange between groundwater and surface water to a fluvial environment where such measurements are very challenging to obtain.  Furthermore, this multi-method approach was applied to a pressing hydroecological question regarding the potential tie between groundwater discharge and patchy occurrence of the endangered dwarf wedgemussel species.  The range of results and the sometimes lack of agreement of results are further evidence and a good example of the difficulty associated with conducting research in such a setting.  One major conclusion is that all of these methods are necessary to reach comprehensive and defendable conclusions regarding groundwater discharge in relation to mussel beds.  Lastly, as to the lack of inter-method comparison and discussion of results, the reviewer is correct.  We will provide a much more thorough discussion of methods comparisons, including reasons for lack of concordance in some cases, in the discussion section.

**Responses to specific Reviewer #1 comments**

Reviewer 1:  Figure 2 – the text (lines 172-175) refers to locations were seepage meters were installed.
However, the figure itself only mentions seeps, which I take to mean actual locations where seepage was observed. Moreover, on line 175, the authors mention locations S1 and S2, but no such locations are shown on the figure.

Authors' reply: The figure was increased in size to allow labeling of all locations where seepage meters and piezometers were installed, including the two locations S1 and S3 situated within the spring area at Site 2.

Reviewer 1: Line 181 – the opening paragraph indicates that pressure is measured in streambed
piezometers. Actually, this is incorrect. Head is measured. The authors say this later
in the paragraph (line 187) – vertical head differences.

Authors' reply: The reviewer is correct. Although the sensor actually measures pressure in kPa, we convert that value to hydraulic head and should have described the measurement as such. We have revised to write hydraulic head instead of pressure.

Reviewer 1: Figure 4 – The locations of the seepage meters and 1D temperature sensors cannot
be seen on this figure. Some other symbol color (blue?) is needed to differentiate from
the conductivity. However, I don't think that the instrument locations should be shown
in this figure anyway – see my next comment. The EMI results are quite separate from
the thermal results and are not even discussed together.

Authors' reply: We have changed the symbols to more clearly indicate the locations of seepage meters and temperature profilers. We had not realized the extent to which the symbols become lost when reduced to page size. As to whether to include the symbols on the figure, we prefer to include the locations of these sensors. They provide a useful indication of the relative positioning of measurements of seepage discharge with an indication of underlying geology that may be related to seepage discharge.

Reviewer 1: Figure 2 should be modified or a new figure included to show the locations of measurements
of seepage (numbered according to Table 1). Note, locations M4 and M5
are also referred to in the text (line 260) but are not shown anywhere. This figure could
also show the 1D sensor locations.

Authors' reply: We have revised the figure to show the locations of all seepage-meter and piezometer locations. We chose to present the 1-d sensor locations in Figure 4 only to reduce clutter in Figure 2.

Reviewer 1: Line 314-315, and Line 323-324 – stronger justification is needed about the crosssectional
area. Why 10 to 20 m. Where is the evidence of this? I realize that a range
is given, but thickness seems like a rather important parameter.

Authors' reply: We agree that thickness is an important parameter. Based on our observations of outcrops and reports of surficial geology and soil surveys of the area, our best estimate of the vertical section that readily transmits water is 10 to 20 m. We have added two citations from which we based our best estimate.

**Reviewer #1 Technical corrections**

Authors' reply:  We thank the reviewer for making editorial corrections to the manuscript and we have accepted all suggestions and have revises accordingly, with one exception.  The reviewer suggested the following:

Line 160 – some explanation of how A was assessed is needed. How large was the vertical plane? Given that the volumetric seepage rate ($Q$) is being compared between sites, it might be better to compare the flux.

We agree with the reviewer and have already revised the manuscript to indicate how we arrived at a vertical thickness of 10 to 20 m, as per the reviewer's earlier suggestion.  However, because we compare $Q$ based on the Darcy equation with $Q$ based on discharge from springs and seeps, we prefer to present the volumetric flow here in terms of $Q$ rather than $q$.

**Responses to Anonymous Reviewer #2 general comments**

Anonymous reviewer #2 also was concerned about the lack of clarity regarding all datasets pointing to the conclusion that groundwater discharge is correlated with presence of endangered mussels.  This compelled us to better and more clearly explain that it took the full suite of methods and lines of evidence to reach this conclusion.  We have revised the text to more emphatically indicate that mussels are present in areas of greater groundwater discharge in general, but they do not appear to be located specifically at points of more rapid or focused discharge.

We also were pleased by the success of simply making manual temperature measurements of the riverbed and yet we failed to compare results from manual bed-temperature surveys compared with the much more expensive and sophisticated fiber-optic measurements.  We will point out this comparison and include likely reasons for the differences in results.

**Responses to specific Reviewer #2 comments**

Reviewer 2:  L133: The piezometers where groundwater samples were collected are not shown on Figure 2?

We have revised the figure to allow us to add labels for specific seepage meter and piezometer locations.

Reviewer 2:  L141: Suspended sediment. That suggests to me a very high discharge. Were they found at all three sites?

Authors' reply:  Indeed it does.  Suspended sediment, sometimes described as "boiling sand", is indicative of a seepage rate large enough that sediments are lifted off of the bed.  This condition was

observed at Sites 1 and 3.  At Site 2, the rapid groundwater discharge occurred just above the river surface and visibly flowed downslope to the river.  At all sites, zones of strong groundwater discharge were spatially discrete at the centimeter scale.  We now point out the relevance of this important feature that provides strong evidence of focused groundwater discharge in these areas.

Reviewer 2:  L175: Are these places with suspended sediments?

Authors' reply:  Although the spring area had attributes of focused and rapid groundwater discharge, namely very cold water and soft sediments, we did not observe suspended sediment within the spring area.  Although groundwater discharge was obviously prodigious, it was not rapid enough to suspend sediments at the submerged portion of the riverbed.  We have added text in the Discussion section (lines 478-482) that emphasizes this feature.

Reviewer 2:  L310-318: You estimate a rather low K value (0.063 m/d), which I think represents silt more than sand. You call it different things; sandy sediments, silty sand, sand and fractured shallow sandstone, in this section. Maybe it is not so important what it is called, more so that this diffuse discharge is low? At least, here I am left with the impression that it is not the diffuse groundwater discharge that sustains DWM populations?

Authors' reply:  The sand description was based on sediment removed during well installation.  The silty sand description was based on the hydraulic-conductivity value from the slug test.  The fractured sandstone was based on the observation that the surface of the sandstone bedrock typically is relatively more fractured.  We had indicated that sediment was sandy during well installation and that bedrock was composed primarily of sandstone in an earlier version of the manuscript but we were clearly overzealous in paring the manuscript for submission to HESS.  We can easily replace that information for clarity.  However, the bottom line, as the reviewer surmised, is that the sediments adjacent to the river are relatively fine, indicating that diffuse groundwater discharge like is not particularly large in these areas, suggesting that focused, more rapid discharge is more likely related to presence of this endangered species of mussel.  This point is now emphasized more clearly in the Discussion section.

Reviewer 2:  L319-327: The same goes for site 3 with even lower discharge. Maybe the last sentence is important in this context and for the whole study. That it is not slow, diffuse groundwater discharge that is responsible for sustaining DWM populations, but, rather the preferential flow paths.

Authors' reply:  The reviewer is correct in assuming that diffuse GW discharge indeed likely is less influential in determining the presence/absence of these endangered mussels.  We have stated this point more emphatically in the Discussion section.

Reviewer 2:  L337-339: Unclear.

Authors' reply:  We may have gotten a bit down in the weeds with our description of these flow reversals.  Our main point was that they were short-lived and that, even only 18 hours after peak river

stage, when river stage was still 0.75 m higher than the pre-event stage, the normal upward hydraulic gradient had already become established.  We have revised the text here to better emphasize this point.

Reviewer 2:  L344-345: These "median" values are also the first and last entries in Table 1 – is that just a coincidence?

Authors' reply:  How interesting.  That indeed is purely coincidental.

Reviewer 2:  L345: I find that there are other sites with "consistently" upward seepage, so does consistently mainly refer to "substantial", i.e., they show high fluxes and does not refer to "direction"?

Authors' reply:  A good point.  We should have stated that this reach had seepage that was consistently upward at substantial rates.  Other reaches had a preponderance of upward seepage values but many were small with values well less than 1 cm/d.  Seepage at all locations but one at the M reach of Site 3 was upward at rates of 3.5 to nearly 27 cm/d.  We have revised to write "The only reach where seepage was substantially upward was the M reach at Site 3, where upward seepage at 4 of 5 locations was greater than 3 cm/d."

Reviewer 2:  L367: Is it fair to say that this indicates a greater potential for seepage at N locations, which goes against your conclusion?

Authors' reply:  We don't think so.  Although median Kv at the N reach of Site 2 was larger than at the M reach, both values were essentially the same (1.3 and 0.5), given the uncertainty associated with determining K.  The much larger Kv at the N reach at Site 3 indicates that water can flow more easily through those sediments, but the median vertical hydraulic gradient was essentially zero at the N reach, indicating a small potential for seepage.

Reviewer 2:  L374: With my experience some of the fluxes are not just moderate, but also high.

Authors' reply:  The largest value was just over 84 cm/d, which indeed is a large number for lake settings.  However, numerous studies have reported seepage in other fluvial settings that is much faster, with values well over 100 cm/d.  Given the coarseness of the bed and the relatively fast river current along the Site 3 N reach, we were actually expecting to measure faster seepage than we did in response to hyporheic exchange.  That is why we chose to call seepage at that reach moderate rather than fast.

Reviewer 2:  L382: How is this error bound explained? Maybe you should also put an error bound on the seepage meters?

Authors' reply:  The temperature-based flux expected error range was determined for similar sediments using a Monte Carlo analysis where sediment thermal properties were varied simultaneously within expected ranges. This previous work (Briggs et al 2012b) is now cited in section 3.2.5 where we present

methods associated with vertical temperature profilers. Errors associated with seepage-meter measurements depend on the setting in which measurements are conducted. Figure 7 presents median values along with maximum and minimum values at each location. We think this figure nicely indicates the relative uncertainty associated with seepage-meter measurements at each location. We have also added a sentence in section 3.2.3 stating that the range of seepage measurements at specific sites incorporates both measurement uncertainty and temporal variability.

Reviewer 2: L423-425: I am not entirely convinced about this conclusion. To me your data and observations collectively suggest that it is not the diffuse groundwater discharge (rather low), but known seeps and unknown preferential flow paths that are related to DWM occurrence. (1) Seeps, I agree. Your observations support this. (2) fluxes measured by seepage meters. I am not quite convinced as you measure positive/negative fluxes in all places and by the fact that they do not compare well with fluxes estimated from temperature profiling. (3) Hydraulic gradients. Maybe not the best measure as the flux will depend on Kv. And, Kv was higher at N sites?

Authors' reply: We agree that observed seeps and springs are probably the strongest evidence that groundwater discharge is much greater in areas populated by endangered mussels. We also think seepage-meter data are fairly convincing. Seepage meter fluxes, although clearly influenced by hyporheic exchange, collectively indicate greater upward seepage at M reaches than at N reaches. Median seepage at the M reach at Site 2 was more than double the N reach value and median seepage at the M reach at Site 3 was the largest of all, while the median seepage at the corresponding N reach was slightly downward. The "noise" attributed to hyporheic exchange in other fluvial settings also has been dealt with in other studies by averaging values across sites to separate the larger-scale groundwater exchange from the hyporheic exchange. We agree that basing seepage rates on hydraulic gradients in these settings is suspect as the hydraulic gradients commonly are very small and difficult to measure. Seepage determined from vertical-temperature profiling also can be problematic due to non-vertical flow associated with hyporheic exchange. So it is not surprising that temperature-based seepage did not always compare with physically based seepage. We have revised and embellished our discussion of these issues the previously appeared on lines 466-476 to more strongly emphasize these points.

Reviewer 2: L434-435: Which brings me to this conclusion, which I am not sure I fully understand or agree with. On the one hand, you argue that DWM rely on "substantial" discharge, which I read as high fluxes; on the other hand, that DWM do not rely on focused (=high?) discharge.

Authors' reply: The point of lines 434-435 is that individual animals do not seem to be located precisely at a point of focused groundwater discharge but instead rely on one or numerous points of focused groundwater discharge that occur nearby. We have revised this text to make this point more clearly, and we now write, "DWM do not require focused groundwater discharge precisely where they are located, but instead rely on the existence of substantial groundwater discharge within or just upstream of their populated area."

Reviewer 2: L441-442: Upward seepage .. is primarily the result of groundwater discharge. Is that not obvious, or are you referring to the possibility of hyporheic flow?

Authors' reply: This text was comparing conditions at the Delaware River with seepage rates reported in another study. Taken out of context (or evidently even in context), the sentence is confusing. We have revised to write "The net upward seepage at DWM sites in the Delaware River, although clearly influenced by hyporheic exchange, is primarily the result of groundwater discharge as evidenced by substantially faster reach-averaged upward seepage and also colder water along M reaches relative to N reaches." This now appears in lines 463-466.

Reviewer 2: L468-470: Why will the methods produce two different results because of a cobble-bed river?

Authors' reply: This is a very good question and one that is often difficult to understand. Seepage meters measure flux across the bed no matter the actual vector direction of the flow. Water can be flowing primarily horizontally beneath the bed and a seepage meter will measure whatever water crosses the sediment-water interface beneath the meter. However, hydraulic gradients and seepage based on vertical temperature profiling assumes that flow through the sediment is vertical. Furthermore, they integrate the flow based on the positioning of the well screen or temperature sensors, which may be well beneath the bed and influenced by subsurface flowpaths that are not related to flow at the bed surface at that specific location. Others papers also have observed this feature that is a characteristic of some hyporheic settings. We clearly did not properly explain this situation. We have revised the text here to hopefully convey this concept more clearly: "Data indicating flow in opposite directions across the riverbed are initially puzzling (Table 2). Some of the discordant data may be attributed to measurement error. Vertical hydraulic gradients at several in-river piezometers were very small (and difficult to measure), as were some of the seepage rates measured with seepage meters. Furthermore, hyporheic flowpaths in substantially heterogeneous and highly transmissive sediment, a common situation in a cobble-bed river, is predominantly horizontal with small upward and downward flow components. Because piezometers and vertical temperature profilers are installed vertically, interpretations of hydraulic gradient and seepage assume vertical flow through the sediments, often a poor assumption in hyporheic settings. It is not uncommon for seepage meters to indicate upward flow while hydraulic gradients indicate downward flow (Rosenberry and Pitlick, 2009; Rosenberry et al., 2012; Angermann et al., 2012; Käser et al., 2009). Locations with discordant data indicate flow across the sediment-water interface was largely driven by hyporheic processes, which is superimposed on larger-scale groundwater discharge patterns (Rosenberry et al., 2012)."

Reviewer 2: L471: Now you argue that hyporheic flow can dominate, see comments above?

Authors' reply: Yes, particularly at the N reach of Site 3 where cobbles and K values were largest.

Reviewer 2:  L490: Are you then saying that discharge cannot happen uniformly/diffusively, but must occur as springs, focused flows, through preferential flow paths?

Authors' reply:  Not quite.  We are saying that hyporheic exchange greatly affects measurements and interpretations of larger-scale groundwater discharge to the river.  Perhaps due to hyporheic exchange, or perhaps due to underlying geologic heterogeneity, focused groundwater discharge also occurs, likely in addition to diffuse groundwater discharge.  It seems that DWM are located in areas where locations or amounts of focused groundwater discharge are particularly numerous or large.

**Reviewer #2 Technical corrections**

We agree with all of the technical/editorial suggested by Reviewer #2 and have made all suggested changes to text and figures.

[revised manuscript text omitted]

818
819    Figure 1
820

[Figure]

[Figure]

823
824    Figure 2

[Figure]

825
826    Figure 3

[Figure]

832

[Figure]

833
834    Figure 4
835
836
837

[Figure]

[Figure]

838
839  Figure 5

[Figure]

855
856    Figure 6

[Figure]

865
866    Figure 7

876

877

878
879 Figure 8
880

[Figure]

881
882    Figure 9